# Similarity-Dissimilarity Loss for Multi-label Supervised Contrastive Learning

**Guangming Huang**                                                                   *guangming.huang22@gmail.com*
*University of Essex*

**Yunfei Long**                                                                            *yunfei.long@qmul.ac.uk*
*Queen Mary University of London*

**Cunjin Luo**                                                                             *cunjin.luo@essex.ac.uk*
*University of Essex*

**Reviewed on OpenReview:** *https://openreview.net/forum?id=W445zcqThv*

## Abstract

Supervised contrastive learning has achieved remarkable success by leveraging label information; however, determining positive samples in multi-label scenarios remains a critical challenge. In multi-label supervised contrastive learning (MSCL), multi-label relations are not yet fully defined, leading to ambiguity in identifying positive samples and formulating contrastive loss functions to construct the representation space. To address these challenges, we: (i) systematically formulate multi-label relations in MSCL, (ii) propose a novel *Similarity-Dissimilarity Loss*, which dynamically re-weights samples based on similarity and dissimilarity factors, (iii) further provide theoretically grounded proofs for our method through rigorous mathematical analysis that supports the formulation and effectiveness, and (iv) offer a unified form and paradigm for both single-label and multi-label supervised contrastive loss. We conduct experiments on both image and text modalities and further extend the evaluation to the medical domain. The results show that our method consistently outperforms baselines in comprehensive evaluations, demonstrating its effectiveness and robustness.

## 1 Introduction

While supervised contrastive learning effectively leverages label information to achieve promising results in single-label scenarios (Khosla et al., 2020; Zhang et al., 2022; Lin et al., 2023), identifying positive samples in multi-label supervised contrastive learning (MSCL) remains a fundamental challenge (Zhang & Wu, 2024). For example, consider a set of images containing cats and puppies, wherein an anchor image depicts a cat; in the single-label paradigm, positive and negative instances can be unambiguously delineated based on their corresponding taxonomic annotations. Conversely, MSCL introduces inherent classification ambiguity when determining whether an image containing both cats and puppies should be designated as a positive or negative sample in relation to the anchor.

A critical question arises: *Should a sample be considered positive if its label set partially overlaps with or exactly matches that of the anchor?* Currently, three principal strategies[1] exist for identifying positive samples in multi-label scenarios: (i) *ALL* considers a sample positive only if its label set matches exactly; (ii) *ANY* treats samples with any overlapping class as positive; and (iii) *MulSupCon* (Zhang & Wu, 2024) conceptually aligns with the ANY approach but treats each label independently, thereby generating multiple distinct positive sets for individual anchors.

---

[1]We discuss other related methods in Appendix F.

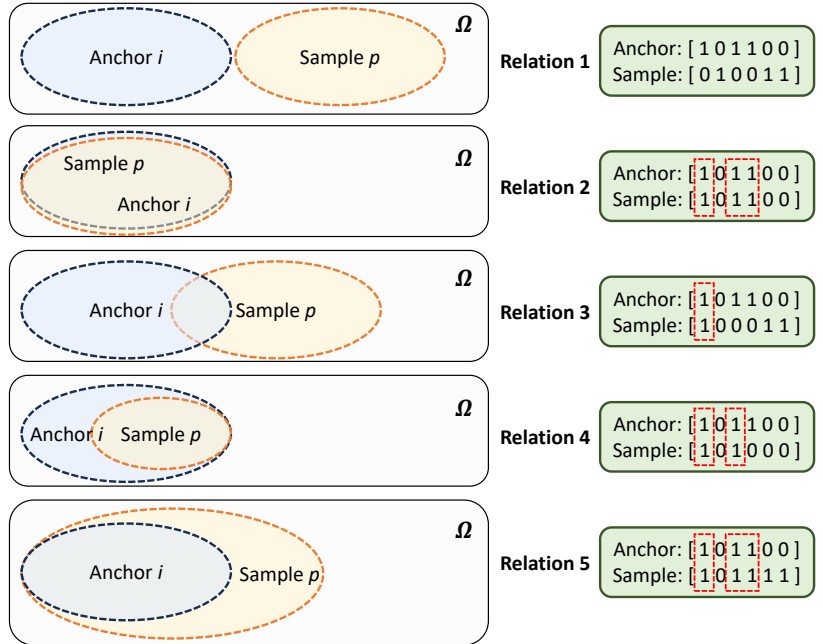

Figure 1: Five distinct multi-label relations between samples and a given anchor. $\Omega$ denotes a universe that contains all label entities. Here is an example with five different relations between sample $p$ and anchor $i$, where the labels are represented as one-hot vectors.

However, these methods have inherent limitations, since previous research has overlooked the complicated multi-label relations among samples in MSCL. As illustrated in Figure 1, we introduce five distinct set relations among samples to facilitate a more comprehensive identification of positive sets. The ALL method exclusively considers relation $R_2$ while disregarding the potential contributions of $R_3$, $R_4$ and $R_5$. Furthermore, under long-tailed distributions, when tail samples serve as anchors, the requirement of ALL for exact label matches significantly impedes these tail anchors from identifying adequate positive samples within a limited batch size, potentially degenerating the method to unsupervised contrastive learning in extreme scenarios (Chen et al., 2020; He et al., 2020; Zhang et al., 2023). Conversely, both ANY and MulSupCon approaches treat relations $R_2$, $R_3$, $R_4$, and $R_5$ identically with equivalent weights in contrastive loss functions, which constitutes a suboptimal approach given the inherent differences among these relations. A detailed mathematical analysis of the limitations for these methods is presented in Section 2 and additional discussion of related work in Appendix F.

To address the aforementioned ambiguities of inter-label relations in MSCL, we define multi-label relations and propose a novel contrastive loss function. *Importantly, this study focuses exclusively on inter-label relations in MSCL, rather than on intra-label or hierarchical relations. Accordingly, the term* **relation** *in this paper refers specifically to inter-label relations.* Our main contributions are summarized as follows:

- We introduce the concept of multi-label relations in MSCL and propose the Similarity-Dissimilarity Loss, which dynamically re-weights positive pairs based on these relations.

- We establish theoretical foundations for the proposed framework by analyzing vector similarity under label equivalence, deriving bounds for the similarity-dissimilarity weighting factors, and characterizing how the proposed loss differentiates distinct multi-label relations.

- We demonstrate that the proposed loss offers a unified formulation that gracefully reduces to standard SupCon (Khosla et al., 2020) in single-label scenarios.

- We validate our approach across image, text, and medical benchmarks, achieving consistent improvements and state-of-the-art performance on the MIMIC datasets.

## 2 Methods

### 2.1 Preliminaries

Given a batch of $N$ randomly sample/label pairs, $\{(\boldsymbol{x}_i, \boldsymbol{y}_i)\}_{i=1,\ldots,N}$, where $\boldsymbol{x}_i$ denotes the $i$-th sample and $\boldsymbol{y}_i$ its corresponding labels. Here, $\boldsymbol{y}_i = \{y_i^{(l)}\}_{l=1,\ldots,L}$ represents the label set of sample $i$, where $y_i^{(l)}$ denotes the $l$-th label of sample $i$ and $L$ is the total number of labels for sample $i$. After data augmentation, the training batch consists of $2N$ pairs, $\{\tilde{\boldsymbol{x}}_j, \tilde{\boldsymbol{y}}_j\}_{j=1,\ldots,2N}$, where $\tilde{\boldsymbol{x}}_{2i}$ and $\tilde{\boldsymbol{x}}_{2i-1}$ are two random augmentations of $\boldsymbol{x}_i$ ($i = 1, \ldots, N$) and $\tilde{\boldsymbol{y}}_{2i-1} = \tilde{\boldsymbol{y}}_{2i} = \boldsymbol{y}_i$. For brevity, we refer to this collection of $2N$ augmented samples as a "batch" (Khosla et al., 2020).

### 2.2 Multi-label Supervised Contrastive Loss

In MSCL, the formulation of supervised contrastive loss varies depending on the strategies employed for determining positive samples relative to a given anchor. Let $i \in \mathcal{I} = \{1, \ldots, 2N\}$ denote the index of an arbitrary augmented sample. For the ALL strategy, the positive set is defined as follows:

$$\mathcal{P}(i) = \{p \in \mathcal{A}(i) | \forall p, \tilde{\boldsymbol{y}}_p = \tilde{\boldsymbol{y}}_i\} \tag{1}$$

where $\mathcal{A}(i) \equiv I \setminus \{i\}$ [2].

Subsequently, the positive set for the ANY strategy is defined as follows:

$$\mathcal{P}(i) = \{p \in \mathcal{A}(i) | \forall p, \tilde{\boldsymbol{y}}_p \cap \tilde{\boldsymbol{y}}_i \neq \varnothing\} \tag{2}$$

In MSCL, the form of the contrastive loss function for ALL and ANY is identical. For each anchor $i$, the loss function is formulated as follows:

$$\mathcal{L}_i = \frac{-1}{|\mathcal{P}(i)|} \sum_{p \in \mathcal{P}(i)} \log \frac{\exp(\boldsymbol{z}_i \cdot \boldsymbol{z}_p / \tau)}{\sum_{a \in \mathcal{A}(i)} \exp(\boldsymbol{z}_i \cdot \boldsymbol{z}_a / \tau)} \tag{3}$$

Here, $\tau \in \mathbb{R}^+$ represents a positive scalar temperature parameter (Chen et al., 2020), while $\boldsymbol{z}_k = Proj(Enc(\tilde{\boldsymbol{x}}_k)) \in \mathbb{R}^{D_P}$ denotes the projected encoded representation (Khosla et al., 2020).

For a given batch of samples, the loss function is formulated as:

$$\mathcal{L} = \sum_{i \in I} \mathcal{L}_i \tag{4}$$

Zhang & Wu (2024) propose an approach that considers each label $\tilde{y}_i^{(l)}$ independently, forming multiple positive sets for a given anchor sample $i$. For each label $\tilde{y}_i^{(l)} \in \tilde{\boldsymbol{y}}_i$, the positive set for MulSupCon is defined as:

$$\mathcal{P}(i) = \{p \in \mathcal{A}(i) | \forall p, \tilde{y}_p^{(l)} \in \tilde{\boldsymbol{y}}_i\} \tag{5}$$

For each anchor $i$, the multi-label supervised contrastive loss for MulSupCon is represented as follows (Zhang & Wu, 2024):

$$\mathcal{L}_i^{\text{mul}} = \sum_{\tilde{y}_p^{(l)} \in \tilde{\boldsymbol{y}}_i} \frac{-1}{|\mathcal{P}(i)|} \sum_{p \in \mathcal{P}(i)} \log \frac{\exp(\boldsymbol{z}_i \cdot \boldsymbol{z}_p / \tau)}{\sum_{a \in \mathcal{A}(i)} \exp(\boldsymbol{z}_i \cdot \boldsymbol{z}_a / \tau)} \tag{6}$$

For a given batch of samples, the loss function is formulated as:

$$\mathcal{L}^{\text{mul}} = \frac{1}{\sum_i |\tilde{\boldsymbol{y}}_i|} \sum_{i \in I} \mathcal{L}_i^{\text{mul}} \tag{7}$$

---

[2] In contrastive learning, sample $i$ is the anchor and should be excluded from the positive set.

### 2.3 Multi-label Relations

As illustrated in Figure 1, we denote each *Relation* as $R$, where, e.g., $R_1$ stands for *Relation 1*. The subscripted notation $p_j$ signifies that sample $p$ corresponds to the $j$-th relation.

Let $\Omega$ denote a universal set containing all possible label entities. For any anchor $i$ and sample $p$, let $\mathcal{S}$ and $\mathcal{T}$ represent their respective label sets. The five fundamental multi-label relations are defined as follows:

$$R_1 : \mathcal{S} \cap \mathcal{T} = \varnothing \tag{8}$$
$$R_2 : \mathcal{S} = \mathcal{T} \tag{9}$$
$$R_3 : \mathcal{S} \cap \mathcal{T} \neq \varnothing, \mathcal{S} \nsubseteq \mathcal{T}, \mathcal{T} \nsubseteq \mathcal{S} \tag{10}$$
$$R_4 : \mathcal{S} \supsetneq \mathcal{T} \tag{11}$$
$$R_5 : \mathcal{S} \subsetneq \mathcal{T} \tag{12}$$

Based on these relational definitions, we present a theoretical analysis of the limitations inherent in the ALL, ANY, and MulSupCon methods, illustrated via an example in Figure 1.

In ALL, the optimization process aims to align with the mean representation of samples sharing identical label sets (Zhang & Wu, 2024). As the example that is demonstrated in Figure 1, for a given anchor $i$, the positive set of ALL is:

$$\mathcal{P}(i) = \{p_2\}$$

In ALL, the sample $p_j$ in $R_2$ is designated as a positive sample, while those in relations $R_3$, $R_4$ and $R_5$ are excluded from consideration. Specifically, despite their semantic similarity to anchor $i$ due to overlapping labels, the feature representations of samples $p_j$ where $j \in 3, 4, 5$ are forced away from the anchor in the embedding space, as they are treated as negative examples in the contrastive learning paradigm. Consequently, the restricted size of the positive set $|\mathcal{P}(i)|$ results in a mean representation susceptible to statistical variance. Furthermore, the ALL method may inadvertently treat semantically related samples as negative instances in certain scenarios.

**Lemma 1.** *(Vector Similarity Under Label Equivalence). Let $i$ be an anchor and $p$ be any sample in the feature space, where $\tilde{\boldsymbol{y}}_i, \tilde{\boldsymbol{y}}_p \in \mathbb{R}^d$ denote their respective label vectors. If $\tilde{\boldsymbol{y}}_p = \tilde{\boldsymbol{y}}_i$, then under the contrastive learning framework (Chen et al., 2020), their corresponding projected representations $\boldsymbol{z}_i, \boldsymbol{z}_p \in \mathbb{R}^m$ satisfy $\boldsymbol{z}_i \simeq \boldsymbol{z}_p$.*

*Proof.* See Appendix B.1. □

**Interpretation.** Lemma 1 formalizes the intuition that exact label-set equivalence provides a reliable positive relation: when two samples share identical labels, their projected representations are expected to be close under the contrastive learning objective. This provides a theoretical justification for treating $R_2$ as a high-confidence positive relation.

As per ANY's definition, the positive set of the example in Figure 1 is:

$$\mathcal{P}(i) = \{p_2, p_3, p_4, p_5\}$$

By applying Lemma 1, the corresponding loss terms in Equation 3 for samples in different relations exhibit approximate equality:

$$\mathcal{L}(R_2) \approx \mathcal{L}(R_3) \approx \mathcal{L}(R_4) \approx \mathcal{L}(R_5)^3$$

It is evident that $R_2$, $R_3$, $R_4$ and $R_5$ represent fundamentally distinct relations, each characterized by different labels and semantic information. However, ANY fails to differentiate these subtle label hierarchies,

---

[3]The approximation notation is used instead of equality due to vector similarity in Lemma 1 and the inherent uncertainty in deep learning's non-linear transformations.

introducing substantial semantic ambiguity. Moreover, in scenarios where samples predominantly share common classes, the averaging mechanism disproportionately emphasizes these shared classes while diminishing the significance of distinctive features (Zhang & Wu, 2024).

MulSupCon employs a positive sample identification mechanism analogous to ANY; samples $p_j$, where $j \in 3, 4, 5$ are designated as positive instances. However, MulSupCon distinguishes itself by evaluating each label individually and forming multiple positive sets for a single anchor sample. This approach aggregates positive samples based on the number of overlapping labels between the positive samples and the anchor, thereby expanding the space of positive sets:

$$\mathcal{P}(i) = \{p_2, p_2, p_2, p_3, p_4, p_4, p_5, p_5, p_5\}$$

Subsequently, the loss terms for $p_j$ in Equation 6 are as follows by Lemma 1:

$$\mathcal{L}(R_2) \approx \mathcal{L}(R_5) \neq \mathcal{L}(R_3) \neq \mathcal{L}(R_4)$$

For this example (see Figure 1), MulSupCon successfully discriminates $R_3$ and $R_4$ from $R_2$ and $R_5$; however, it fails to establish a distinction between $R_2$ and $R_5$. This limitation arises primarily because MulSupCon exclusively considers the overlapping regions (*Similarity* [4]) between anchor $i$ and sample $p$ (i.e., the intersection of sets $\mathcal{S}$ and $\mathcal{T}$), while disregarding the complementary non-intersecting domains (*Dissimilarity* [5]). That is to say, the similarity between positive samples and anchors is considered, yet dissimilarity is not, although it provides critical information for representation learning in MSCL.

Leveraging the proposed multi-label relations, our theoretical analysis systematically elucidates the limitations of existing methods and establishes a rigorous foundation for investigating the profound exploration of concepts of similarity and dissimilarity, and the design of a contrastive loss function.

### 2.4 Similarity-Dissimilarity Loss

To address the aforementioned challenges, we introduce the concepts of similarity and dissimilarity based on set-theoretic relations: (i) As depicted in Figure 1, *Similarity* represents the intersection of sets (i.e., $\mathcal{S} \cap \mathcal{T}$), and (ii) we define *Dissimilarity* as the set difference between $\mathcal{T}$ and the intersection $\mathcal{S} \cap \mathcal{T}$ with respect to sample $p$ (i.e., $\mathcal{T} \setminus \mathcal{S} \cap \mathcal{T}$). For each anchor $i$, we formulate the Similarity-Dissimilarity Loss as:

$$\mathcal{L}_i^{\text{our}} = \frac{-1}{|\mathcal{P}(i)|} \sum_{p \in \mathcal{P}(i)} \log \frac{\mathcal{K}_{i,p}^s \mathcal{K}_{i,p}^d \exp(\boldsymbol{z}_i \cdot \boldsymbol{z}_p / \tau)}{\sum_{a \in \mathcal{A}(i)} \exp(\boldsymbol{z}_i \cdot \boldsymbol{z}_a / \tau)} \tag{13}$$

Here, we define $\mathcal{K}_{i,p}^s$ and $\mathcal{K}_{i,p}^d$ that quantify the *Similarity* and *Dissimilarity* factors for a given anchor $i$ and a positive sample $p$, respectively. These factors are formally defined as follows:

$$\mathcal{K}_{i,p}^s = \frac{|\tilde{\boldsymbol{y}}_p^s|}{|\tilde{\boldsymbol{y}}_i|} = \frac{|\mathcal{S} \cap \mathcal{T}|}{|\mathcal{S}|} \tag{14}$$

and

$$\mathcal{K}_{i,p}^d = \frac{1}{1 + |\tilde{\boldsymbol{y}}_p^d|} = \frac{1}{1 + |\mathcal{T} \setminus (\mathcal{S} \cap \mathcal{T})|} \tag{15}$$

where we define the following set-theoretic quantities:

- $|\tilde{\boldsymbol{y}}_i| = |\mathcal{S}|$ denotes the cardinality of the label space $\tilde{\boldsymbol{y}}_i$.

- $|\tilde{\boldsymbol{y}}_p^s| = |\mathcal{S} \cap \mathcal{T}|$ measures the cardinality of the intersection of sets $\mathcal{S}$ and $\mathcal{T}$.

- $|\tilde{\boldsymbol{y}}_p^d| = |\mathcal{T} \setminus (\mathcal{S} \cap \mathcal{T})|$ represents the cardinality of the relative complement with respect to sample $p$.

---

[4]The definition of *Similarity* is introduced in Section 2.4
[5]The definition of *Dissimilarity* is introduced in Section 2.4

The product of $\mathcal{K}_{i,p}^s$ and $\mathcal{K}_{i,p}^d$ is termed as *similarity-dissimilarity factor*. Moreover, the following relation holds:

$$|\tilde{\boldsymbol{y}}_p^d| = |\tilde{\boldsymbol{y}}_p| - |\tilde{\boldsymbol{y}}_p^s| \geq 0 \tag{16}$$

where $|\tilde{\boldsymbol{y}}_p|$ represents the cardinality of the label space associated with sample $p$.

*Importantly*, a detailed mathematical explanation of the re-weighting mechanism for the Similarity-Dissimilarity Loss is provided in Section 2.7. Additionally, Appendix A offers an analysis of the temperature scaling hyperparameter $\tau$ (see Appendix A.1), and the rationale for selecting the dissimilarity penalty function (see Appendix A.2).

### 2.4.1 A Unified Formulation of Supervised Contrastive Loss

Specifically, the Similarity-Dissimilarity Loss reduces to Equation 3 when the following conditions are simultaneously satisfied:

$$\begin{cases} |\tilde{\boldsymbol{y}}_i| = |\tilde{\boldsymbol{y}}_p^s| \\ |\tilde{\boldsymbol{y}}_p^d| = 0 \end{cases} \tag{17}$$

Accordingly, our proposed loss function constitutes a generalized form of the basic supervised contrastive loss (see Equation 3). In particular, Equation 3 represents a special case of the Similarity-Dissimilarity Loss. Moreover, our contrastive loss unifies both single-label and multi-label supervised contrastive loss functions within a comprehensive formulation and paradigm.

### 2.5 Case Analysis

Let us examine the behavior of our loss function through a detailed analysis of five distinct relational cases illustrated in Figure 1. Consider the following sequences of cardinalities:

$$\begin{cases} |\tilde{\boldsymbol{y}}_{p_j}^s| = \{0, 3, 1, 2, 3\}_{j=1,2,3,4,5} \\ |\tilde{\boldsymbol{y}}_{p_j}^d| = \{3, 0, 2, 0, 2\}_{j=1,2,3,4,5} \end{cases}$$

Applying these values to Equation 14 and 15, we obtain:

$$\begin{cases} \mathcal{K}_{i,p}^s = \{0, 1, \dfrac{1}{3}, \dfrac{2}{3}, 1\} \\ \mathcal{K}_{i,p}^d = \{\dfrac{1}{4}, 1, \dfrac{1}{3}, 1, \dfrac{1}{3}\} \end{cases}$$

Consequently, the product of these measures yields:

$$\mathcal{K}_{i,p}^s \mathcal{K}_{i,p}^d = \{0, 1, \frac{1}{9}, \frac{2}{3}, \frac{1}{3}\}$$

By Equation 13 and Lemma 1, these distinct relations ($R_2$ through $R_5$) generate unique loss values, establishing the following inequalities:

$$\mathcal{L}(R_2) \neq \mathcal{L}(R_3) \neq \mathcal{L}(R_4) \neq \mathcal{L}(R_5)$$

The proposed loss function effectively discriminates among the five distinct relations through a principled re-weighting mechanism, as formulated in Equation 13, 14, and 15, compared to existing methods in MSCL.

Furthermore, in contrast to MulSupCon, the Similarity-Dissimilarity Loss preserves the cardinality of positive sets while maintaining computational efficiency, as it requires no additional computational overhead. A detailed computational cost analysis for the proposed loss is provided in Appendix C.1.

### 2.6 Theoretical Analysis

The proposed loss function incorporates a weighting mechanism defined by the product of factors $\mathcal{K}^s_{i,p}$ and $\mathcal{K}^d_{i,p}$. By construction, the *similarity-dissimilarity factor*, $\mathcal{K}^s_{i,p}\mathcal{K}^d_{i,p}$, is bounded within the closed interval $[0,1]$ across all possible relational configurations. Consequently, it is formulated as:

$$\mathcal{K}^s_{i,p}\mathcal{K}^d_{i,p} \in [0,1] \tag{18}$$

For notational conciseness, we denote the product of the similarity and dissimilarity factors across the five relations as $\{\mathcal{K}^s_m\mathcal{K}^d_m\}_{m=1,2,3,4,5}$.

**Theorem 1.** *Let $\mathcal{K}^s_m$ and $\mathcal{K}^d_m$ be the Similarity and Dissimilarity operators, respectively, as defined in Eq. (14) and (15). For the case $m = 1$, their product vanishes:*

$$\mathcal{K}^s_m\mathcal{K}^d_m = 0, \quad when\ m = 1 \tag{19}$$

*Proof.* See Appendix B.2. □

**Theorem 2.** *Consider the Similarity operator $\mathcal{K}^s_m$ and Dissimilarity operator $\mathcal{K}^d_m$ as defined in Eq. (14) and (15). For the case $m = 2$, their product equals unity:*

$$\mathcal{K}^s_m\mathcal{K}^d_m = 1, \quad when\ m = 2 \tag{20}$$

*Proof.* See Appendix B.3. □

**Theorem 3.** *Let $\mathcal{K}^s_m$ and $\mathcal{K}^d_m$ be the Similarity and Dissimilarity operators as defined in Eq. (14) and (15), respectively. For $m \in \{3,4,5\}$, their product is strictly bounded between 0 and 1:*

$$0 < \mathcal{K}^s_m\mathcal{K}^d_m < 1 \tag{21}$$

*Proof.* See Appendix B.4. □

Based on Theorem 1, 2, and 3, the product of the weighting factors $\mathcal{K}^s_{i,p}$ and $\mathcal{K}^d_{i,p}$ is bounded within the interval $[0,1]$, aligning with fundamental principles of loss functions and set-theoretic relations. The non-negative lower bound adheres to the essential property of loss functions being strictly positive (LeCun et al., 2015). Given that our proposed loss function generalizes the supervised contrastive loss (Khosla et al., 2020) and incorporates multi-label relation definitions, the upper bound is naturally capped at 1. This mathematical framework demonstrates that the proposed loss dynamically scales the weighting factors within $[0,1]$ to effectively differentiate sample features, providing a rigorous mathematical justification for both its formulation and efficacy.

**Theorem 4.** *Let $i \in \mathcal{I}$ be a fixed anchor sample, and let $p_3, p_4 \in \mathcal{P}(i)$ be positive samples corresponding to relations $R_3$ and $R_4$, respectively. Suppose their label spaces satisfy the cardinality constraint:*

$$|\tilde{\boldsymbol{y}}_{p_3}| = |\tilde{\boldsymbol{y}}_{p_4}| \tag{22}$$

*Then, the product of similarity and dissimilarity operators satisfies the strict inequality:*

$$\mathcal{K}^s_4\mathcal{K}^d_4 > \mathcal{K}^s_3\mathcal{K}^d_3 \tag{23}$$

*Proof.* See Appendix B.5. □

**Theorem 5.** *Let $i \in \mathcal{I}$ be a fixed anchor sample, and let $p_3, p_5 \in \mathcal{P}(i)$ be positive samples corresponding to relations $R_3$ and $R_5$, respectively. Suppose:*

$$|\tilde{\boldsymbol{y}}^d_{p_5}| \leq |\tilde{\boldsymbol{y}}^d_{p_3}| \tag{24}$$

*Then, the product of Similarity and Dissimilarity operators satisfies the strict inequality:*

$$\mathcal{K}^s_5\mathcal{K}^d_5 > \mathcal{K}^s_3\mathcal{K}^d_3 \tag{25}$$

*Proof.* See Appendix B.6. □

Theorem 4 and 5 establish strict dominance relations among relation types $R_3$, $R_4$, and $R_5$. Specifically, they demonstrate that $\mathcal{K}_4^s \mathcal{K}_4^d > \mathcal{K}_3^s \mathcal{K}_3^d$ when $|\tilde{\boldsymbol{y}}_{p_3}| = |\tilde{\boldsymbol{y}}_{p_4}|$ and, $\mathcal{K}_5^s \mathcal{K}_5^d > \mathcal{K}_3^s \mathcal{K}_3^d$ when $|\tilde{\boldsymbol{y}}_{p_5}^d| \leq |\tilde{\boldsymbol{y}}_{p_3}^d|$. These inequalities, proved through rigorous mathematical derivation using set cardinality properties and fundamental principles of real analysis, reveal a well-defined hierarchical structure in the weighting factors. This hierarchical relation ensures that our loss function appropriately modulates the contributions of different relation types during the learning process, providing theoretical guarantees for the effectiveness of our proposed approach in capturing complex relations within the data.

Overall, our theoretical analysis establishes a comprehensive mathematical foundation for the proposed loss function through five key theorems. These theoretical guarantees, derived through rigorous set-theoretic analysis, demonstrate that our loss function effectively modulates the contributions of different relation types while maintaining proper mathematical bounds, thereby providing a solid theoretical foundation for its application in multi-label contrastive learning.

## 2.7 Re-weighting Mechanism

The proposed re-weighting mechanism is designed to assign soft weights to positive pairs in multi-label contrastive learning, ensuring that the contribution of each positive to the objective function reflects the degree of semantic alignment with the anchor. Formally, the weighting factor for a positive sample $p$ associated with anchor $i$ is given by the product

$$w_{i,p} = \mathcal{K}_{i,p}^s \mathcal{K}_{i,p}^d, \quad w_{i,p} \in [0,1],$$

where $\mathcal{K}_{i,p}^s$ captures similarity based on shared labels, and $\mathcal{K}_{i,p}^d$ penalizes mismatches due to additional labels.

The similarity component $\mathcal{K}_{i,p}^s$ is monotonically increasing with respect to the number of shared labels between $i$ and $p$. Let $L(i)$ denote the set of labels associated with sample $i$. Then,

$$\mathcal{K}_{i,p}^s = f\big(|L(i) \cap L(p)|\big),$$

where $f(\cdot)$ is a non-decreasing mapping that promotes stronger alignment when $p$ and $i$ share more labels, thereby reflecting higher semantic agreement.

The dissimilarity component $\mathcal{K}_{i,p}^d$ instead captures the penalty incurred by extraneous labels present in $p$ but absent in $i$. Specifically,

$$\mathcal{K}_{i,p}^d = g\big(|L(p) \setminus L(i)|\big),$$

where $g(\cdot)$ is a monotonically decreasing function, ensuring that positives with many additional labels contribute less, since these labels may correspond to irrelevant or noisy semantics.

The resulting product $w_{i,p}$ quantifies the *relational quality* between $i$ and $p$. Several important cases arise:

1. **Perfect alignment ($R_2$):** If $L(i) = L(p)$, then $\mathcal{K}_{i,p}^s$ is maximized and $\mathcal{K}_{i,p}^d = 1$, yielding $w_{i,p} \approx 1$.

2. **Partial overlap ($R_3$–$R_5$):** If $L(i) \cap L(p) \neq \varnothing$ but $L(i) \neq L(p)$, then $\mathcal{K}_{i,p}^s$ is non-zero but $\mathcal{K}_{i,p}^d < 1$, leading to an intermediate weight $0 < w_{i,p} < 1$.

3. **Disjoint labels ($R_1$):** If $L(i) \cap L(p) = \varnothing$, then $\mathcal{K}_{i,p}^s = 0$ and $w_{i,p} = 0$, ensuring no contribution to the loss.

This formulation provides a principled mechanism for modulating the contrastive signal. High-quality positives, characterized by substantial semantic overlap and minimal extraneous labels, are emphasized in the embedding space. Conversely, positives with excessive label mismatches are downweighted, thereby mitigating the risk of noisy or misleading training signals. Importantly, the continuous nature of $w_{i,p}$ allows the model to smoothly calibrate the pulling force across a spectrum of semantic relationships, rather than making binary inclusion–exclusion decisions.

Such a mechanism is particularly critical under long-tailed and ambiguous multi-label distributions, where the frequency of classes and the degree of label co-occurrence vary significantly. By balancing similarity and mismatch in a mathematically grounded manner, the re-weighting scheme ensures that the learned representations remain both semantically faithful and robust to noise.

## 3 Experiments and Results

We conduct experiments to compare Similarity-Dissimilarity Loss with baseline loss functions in a comprehensive evaluation, considering: (i) Data modality: image and text data; (ii) Domain-specific: general text data (AAPD) and medical domain (MIMIC III and IV); (iii) Data distribution: full setting (extreme long-tailed distribution) and top-50 frequent labels setting; (iv) ICD code versions: ICD-9 and ICD-10, and (v) Models: ResNet-50, RoBERTa-based, Llama-3.1-8B, and PLM-ICD. We provide full details of experimental setup in Appendix C, and additional results and analysis in Appendix D.

### 3.1 Evaluation on Image

The experimental results in Table 1 demonstrate that our proposed loss function outperforms baselines across almost metrics on image datasets, with the sole exception of a lower mAP than Jaccard (Lin et al., 2023) and MSC (Audibert et al., 2024) on PASCAL. The substantial improvements in macro-F1 (Figure 2a) provide compelling evidence that our method demonstrates exceptional efficacy in addressing long-tailed distribution challenges, a capability particularly crucial in multi-label scenarios.

However, on the PASCAL dataset, the method yields only marginal gains due to its low average label cardinality (approximately 1.5), which causes the task to approximate single-label classification; consequently, multi-label loss functions exert limited influence, consistent with prior findings that label cardinality is a significant determinant of MSCL performance (Audibert et al., 2024).

Moreover, Figure 2b shows that PASCAL exhibits much lower variance across methods compared to MS-COCO and NUS-WIDE, indicating that specialized multi-label loss functions become less effective when label cardinality approaches one. This observation supports the theoretical analysis in Section 2, where the Similarity–Dissimilarity Loss reduces to single-label cases (Equation 17). A detailed analysis and discussion of the results on image data is provided in Appendix D.1

Table 1: Results on image datasets (MS-COCO, PASCAL, and NUS-WIDE). We compare our proposed method (Sim-Dissim) with baselines.

| Method | MS-COCO | | | PASCAL | | | NUS-WIDE | | |
|---|---|---|---|---|---|---|---|---|---|
| | micro-F1 | macro-F1 | mAP | micro-F1 | macro-F1 | mAP | micro-F1 | macro-F1 | mAP |
| ALL | 68.93 | 63.32 | 64.11 | 82.53 | 79.87 | 79.32 | 70.25 | 52.84 | 51.35 |
| ANY | 64.80 | 57.37 | 56.90 | 82.31 | 79.65 | 79.15 | 68.42 | 50.65 | 49.28 |
| MulSupCon | 71.33 | 66.25 | 67.69 | 82.75 | 80.26 | 79.58 | 71.88 | 54.36 | 52.47 |
| Sim-Dissim | **73.40** | **70.03** | **69.20** | **83.63** | **81.10** | 79.75 | **73.35** | **57.49** | **56.74** |
| Jaccard | 69.81 | 64.22 | 65.92 | 82.53 | 79.86 | **80.09** | 71.07 | 52.84 | 51.25 |
| Class proto | 71.88 | 67.35 | 68.32 | 81.85 | 79.75 | 78.06 | 71.79 | 56.03 | 52.95 |
| MSC | 71.96 | 67.54 | 68.38 | 82.56 | 80.39 | 80.02 | 72.02 | 56.13 | 52.87 |

### 3.2 Evaluation on Text

We further evaluate our method on general text data, and the results demonstrate that our proposed loss function consistently surpasses baseline methods for both RoBERTa and Llama models across all metrics on the AAPD dataset (See Table 2). In contrast to the significant performance gains observed on image

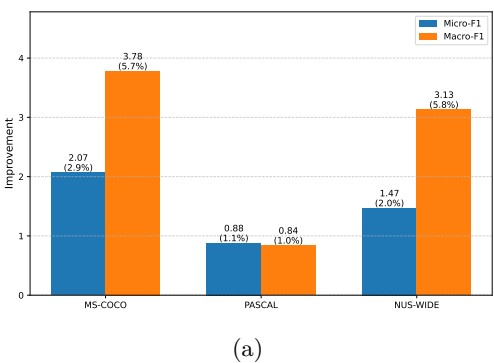
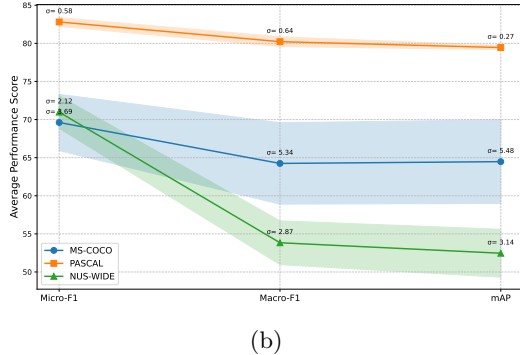

(a)                                                              (b)

Figure 2: (a) Comparison of performance improvements between Similarity-Dissimilarity Loss and MulSup-Con with micro- and macro-F1 metrics. (b) Comparison standard deviation of image datasets on micro-F1, macro-F1 and mAP metrics.

data, Similarity-Dissimilarity Loss achieves more modest enhancements of 0.90/1.79 in micro/macro-F1 scores on RoBERTa, and 0.89/1.84 on Llama. This attenuated performance differential can be attributed to the extensive knowledge already encoded within LLMs through their comprehensive pre-training paradigms (Yang et al., 2024).

Moreover, as illustrated in Figure 3, performance variations of contrastive loss functions for MSCL on both RoBERTa and Llama models are relatively minimal. Specifically, the standard deviations in micro-F1 are 0.80 and 0.79 on RoBERTa and Llama, respectively, while the corresponding standard deviations for macro-F1 metrics are 1.41 and 1.42. Unlike image classification in MSCL paradigm, performance improvements in text classification are predominantly attributable to the intrinsic representational capabilities of model architecture of LLMs. Consequently, while fine-tuning the pre-trained weights of LLMs during the contrastive learning phase can yield marginal performance improvements, this methodological approach demonstrates substantially greater efficacy for visual classification tasks compared to textual classification.

Table 2: Results on AAPD Dataset. We compare our proposed method (Sim-Dissim) with baselines on general text data using RoBERTa-based and Llama-3.1-8B models.

| Method | RoBERTa | | Llama | |
|---|---|---|---|---|
| | Micro-F1 | Macro-F1 | Micro-F1 | Macro-F1 |
| ALL | 73.23 | 59.41 | 74.32 | 60.47 |
| ANY | 72.31 | 58.55 | 73.41 | 59.63 |
| MulSupCon | 73.64 | 60.52 | 74.72 | 61.58 |
| Sim-Dissim | **74.54** | **62.31** | **75.61** | **63.42** |

### 3.3 Evaluation on Medical Domain

Figure 5 reports the gain of our Similarity–Dissimilarity Loss over the strongest baseline for each encoder and dataset setting, showing predominantly positive deltas across both Full and Top-50 label spaces; the improvements are particularly evident on class-balanced metrics (e.g., macro-F1 and macro-AUC), consistent with the claim that dissimilarity-aware re-weighting mitigates noisy positives and benefits tail-label learning. In contrast, micro-level metrics (micro-AUC/micro-F1) tend to be more saturated for strong encoders, yet the proposed method still delivers steady, incremental gains, reflecting improved calibration of the positive set without degrading performance on frequent labels.

Additionally, Figure 4 confirms that these gains translate into stable performance across metrics and encoders rather than isolated wins, indicating that the proposed objective provides a generally better inductive bias for

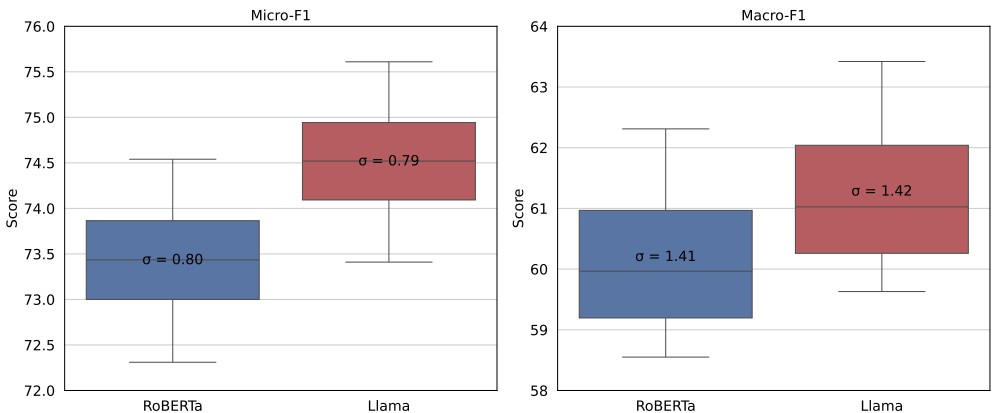

Figure 3: Comparison of RoBERTa and Llama across micro- and macro-F1 on AAPD Dataset.

multi-label relations under both long-tailed (Full) and reduced-label (Top-50) regimes. Moreover, our MSCL framework (PLM-ICD encoder) with Similarity-Dissimilarity Loss outperforms the previous state-of-the-art methods (see Table 3). We provide more experimental results and detailed analysis on medical domain in Appendix D.2.

Table 3: Comparison with previous state-of-the-art medical methods. MRR and AKIL rely on DRG codes, CPT codes, and medications, which are manually annotated for each sample by human coders. These methods are included for reference, although a direct comparison with our approach is not strictly fair. Our method adopts the MSCL framework with Similarity–Dissimilarity Loss and a PLM-ICD encoder.

| Models | AUC | | F1 | | P@8 |
|---|---|---|---|---|---|
| | Macro | Micro | Macro | Micro | |
| CAML (Mullenbach et al., 2018) | 89.5 | 98.6 | 8.8 | 53.8 | 70.9 |
| MSATT-KG (Xie et al., 2019) | 91.0 | 99.2 | 9.0 | 55.3 | 72.8 |
| MSMN (Yuan et al., 2022) | 95.0 | 99.2 | 9.4 | 58.4 | 75.2 |
| KEPTLongformer (Yang et al., 2022) | - | - | 11.8 | 59.9 | 77.1 |
| PLM-ICD (Huang et al., 2022) | 92.6 | 98.9 | 10.4 | 59.8 | 77.1 |
| PLM-CA (Edin et al., 2024) | 91.6 | 98.9 | 8.9 | 57.3 | 74.5 |
| CoRelation (Luo et al., 2024) | 95.2 | 99.2 | 10.2 | 59.2 | 76.2 |
| GKI-ICD (Zhang et al., 2025) | **96.2** | 99.3 | 12.3 | **62.3** | 77.7 |
| Sim-Dissim (ours) | 94.5 | **99.4** | **12.5** | **62.3** | **78.4** |
| MRR (Wang et al., 2024b) | 94.9 | 99.5 | 11.4 | 60.3 | 77.5 |
| AKIL (Wang et al., 2024c) | 94.8 | 99.4 | 11.2 | 60.6 | 78.4 |

## 4  Ablation Study and Robustness Analysis

Table 4 confirms that both factors contribute to performance and that their combination is crucial for obtaining strong and consistent improvements. Although the Similarity-only variant already improves substantially, adding the Dissimilarity factor on top produces a further leap, indicating a synergistic effect that strengthens informative positives while suppressing noisy/partially-related pairs.

To further examine the robustness of the proposed objective, we provide additional analyses in Appendix E. Specifically, we evaluate performance across long-tail label-frequency buckets and study sensitivity to batch size. These results show how the Similarity-Dissimilarity Loss behaves under label imbalance and varying numbers of in-batch positives, these factors that are particularly important in multi-label supervised contrastive learning.

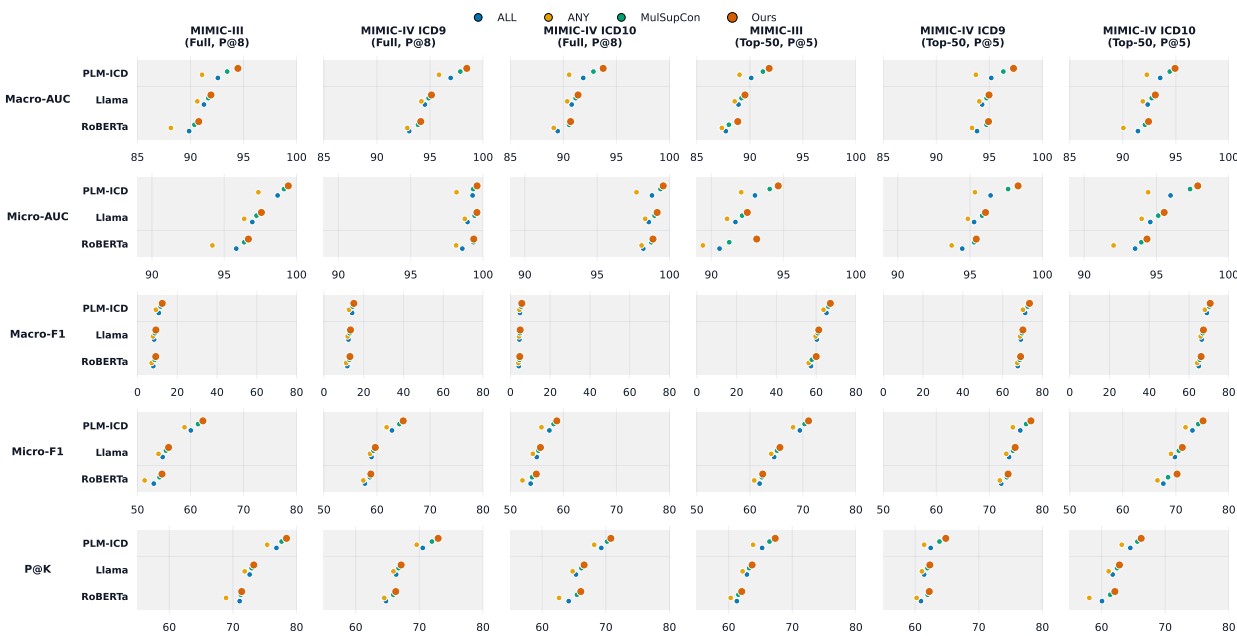

Figure 4: Performance on MIMIC shown as small-multiple dot plots (faceted by dataset and label space; Full vs. Top-50).

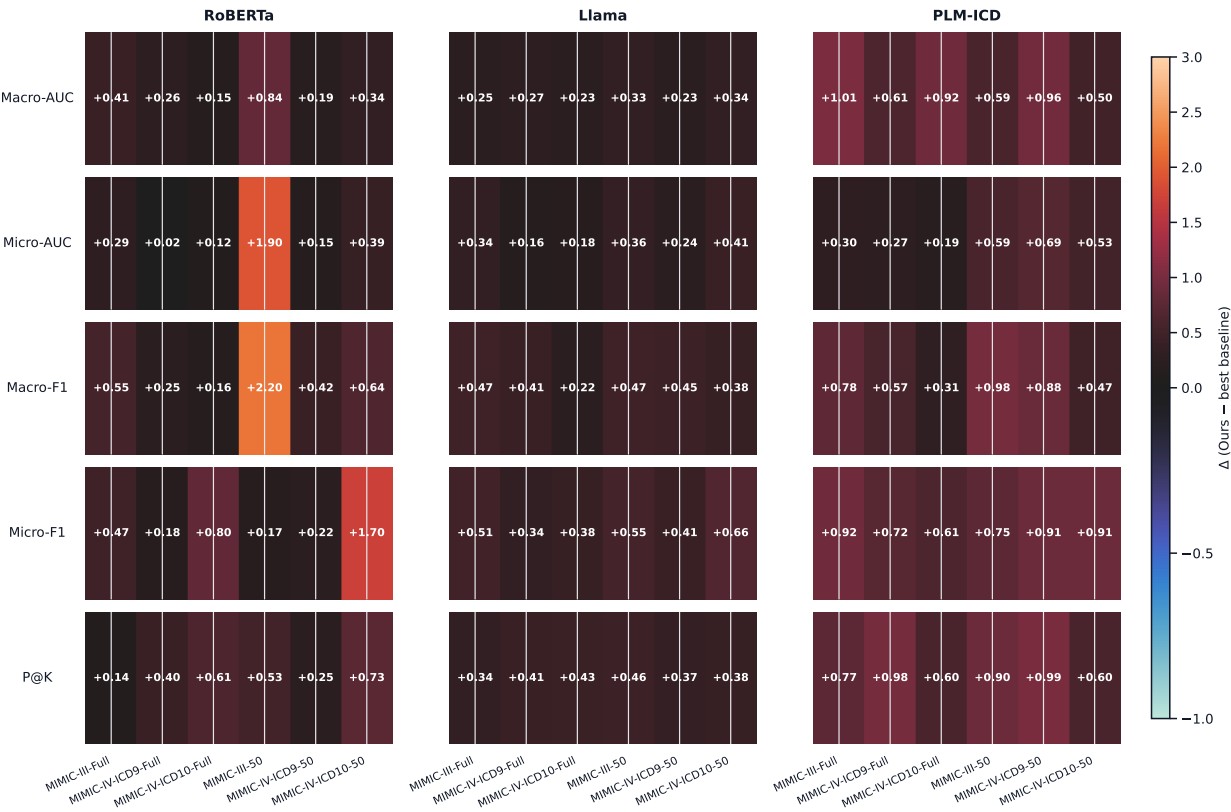

Figure 5: Gain heatstrips on MIMIC comparing our Similarity–Dissimilarity Loss to the strongest baseline for each encoder and setting. Each cell reports $\Delta = \text{Ours} - \max(ALL, ANY, MulSupCon)$ on a given metric; warm colors indicate improvements and cool colors indicate regressions (centered at 0).

Table 4: Ablation study of the comparison of the proposed loss function with and without *Similarity* and *Dissimilarity* factors on image dataset MS-COCO and text dataset AAPD. Encoders and training follow the settings in Appendix C.4

| Factors | | MS-COCO | | AAPD | |
|---|---|---|---|---|---|
| *Similarity* | *Dissimilarity* | Micro-F1 | Macro-F1 | Micro-F1 | Macro-F1 |
| × | × | 64.80 | 57.37 | 72.31 | 58.55 |
| ✓ | × | 67.22 | 62.58 | 73.18 | 59.89 |
| × | ✓ | 65.95 | 59.32 | 72.49 | 58.03 |
| ✓ | ✓ | **73.40** | **70.03** | **74.54** | **62.31** |

## 5 Conclusion

In this paper, by systematically formulating multi-label relations, we establish a principled framework for sample identification in MSCL. Building on this foundation, we propose the *Similarity–Dissimilarity Loss*, which dynamically re-weights samples based on similarity and dissimilarity factors, supported by rigorous theoretical analysis that establishes its mathematical soundness and generalizability to both single-label and multi-label settings. Extensive experiments across image, text, and medical domains demonstrate that our method consistently outperforms strong baselines, achieving state-of-the-art results on the MIMIC datasets. These findings highlight the effectiveness, robustness, and broad applicability of the proposed approach, offering a unified paradigm that advances the study and practice of contrastive learning in complex multi-label scenarios. Moreover, the limitations of this work is discussed in Appendix G.

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

# Appendix

**Contents**

## A   Further Explanation of Similarity-Dissimilarity Loss

### A.1   Explanation of Scaling Temperature

We now explain why we introduce the similarity–dissimilarity weighting by multiplying the factor $\mathcal{K}_{i,p}^s \mathcal{K}_{i,p}^d$ directly into the numerator of the log-softmax loss (Equation 13), rather than scaling the temperature parameter $\tau$ as the formulation is given by

$$\mathcal{L}_i^{\text{our}} = \frac{-1}{|\mathcal{P}(i)|} \sum_{p \in \mathcal{P}(i)} \log \frac{\exp\big((\boldsymbol{z}_i \cdot \boldsymbol{z}_p)/\tau \cdot \mathcal{K}_{i,p}^s \mathcal{K}_{i,p}^d\big)}{\sum_{a \in \mathcal{A}(i)} \exp((\boldsymbol{z}_i \cdot \boldsymbol{z}_a)/\tau)}. \tag{26}$$

**Theoretical Generalization of Supervised Contrastive Loss.** Our formulation (Equation 13) directly generalizes the standard supervised contrastive loss (SupCon) (Khosla et al., 2020), which uniformly weights all positives. When $\mathcal{K}_{i,p}^s = 1$ and $\mathcal{K}_{i,p}^d = 1$ (see Equation 17), it reduces exactly to Equation 3. This structural compatibility would be lost if we instead redefined or modulated the temperature.

**Role and Effectiveness of Temperature in Contrastive Loss.** In the original contrastive learning framework, the temperature hyperparameter $\tau$ controls the sharpness of the softmax distribution over similarity scores. As demonstrated in prior works such as SimCLR (Chen et al., 2020) and further analyzed by Wang & Liu (2021), $\tau$ regulates the concentration level of learned representations: a smaller $\tau$ increases sensitivity to similarity differences, while a larger $\tau$ smooths the gradient signal.

Crucially, temperature $\tau$ acts globally, affecting all pairwise comparisons, including both positive and negative samples. This global effect makes $\tau$ unsuitable for capturing fine-grained, relation-specific weightings such as the similarity–dissimilarity factor, which we intend to apply only to positive samples based on their multi-label relations to the anchor.

From a mathematical perspective, applying the similarity–dissimilarity factor to scaling $\tau$ introduces several issues:

- Global impact on loss structure: Scaling $\tau$ alters contributions of both positive and negative samples, contradicting our objective of selectively modulating the weights of positive pairs based on semantic relations ($R_2$–$R_5$).

- Instability of the loss function: Contrastive loss is highly sensitive to variations in $\tau$. Allowing $\tau$ to vary with each pair according to the similarity–dissimilarity factor may lead to unstable gradients, particularly when the factor approaches 0.

- Theoretical inconsistency: Since $\mathcal{K}_{i,p}^s \mathcal{K}_{i,p}^d \in [0, 1]$ but $\tau \neq 0$ by definition, scaling $\tau$ by this factor is mathematically problematic.

### A.2   Choice of dissimilarity penalty function

**Motivation for Using $\mathcal{K}^d(x) = \frac{1}{1+x}$.**

The dissimilarity factor is intended to softly penalize positive samples that contain additional labels not present in the anchor. Our goals in selecting the function $\mathcal{K}^d(x) = \frac{1}{1+x}$ were to ensure the following properties:

- Boundedness: The function maps dissimilarity to the closed interval $(0, 1]$, which ensures that the contrastive weight remains numerically stable during training.

- Monotonic decay: As dissimilarity increases ($x \uparrow$), the penalty decreases smoothly, discouraging high-variance positives from dominating the loss.

- No hyperparameter sensitivity: The function is parameter-free, making it robust across datasets without the need to tune decay rates or thresholds.

These properties collectively ensure stable gradients, interpretability, and alignment with the goal of penalizing semantically distant positives without completely discarding them.

**Comparison to exponential decay function** $\mathcal{K}^d = \exp(-\alpha x)$:

- While this offers a sharper decay, it introduces a sensitive hyperparameter $\alpha$

- For small values of $x$, the decay may be too aggressive or too mild depending on $\alpha$, making behavior inconsistent across datasets with different label cardinalities.

- It can result in vanishing gradients for large $x$, reducing the contribution of moderately dissimilar positives more than desired.

**Simplicity, Interpretability, and Compatibility.** The chosen form $\mathcal{K}^d(x) = \frac{1}{1+x}$ offers several advantages:

- Simplicity: no tuning required;

- Smoothness: continuous, differentiable, non-zero everywhere. Because $x \geq 0$ and $\frac{1}{1+x}$ is only non-differentiable when $x = -1$.

- Compatibility with our similarity term $K^s$, which also scales linearly in set cardinality.

Together, the similarity-dissimilarity factor $\mathcal{K}^s_{i,p}\mathcal{K}^d_{i,p}$ maintains closed-form interpretability, avoids numerical instability, and allows for clean theoretical analysis, as seen in Theorems 1–5.

# B  Proofs of Lemma and Theorem

## B.1  Proof of Lemma 1

*Proof.* We consider the supervised contrastive loss defined in Equation 3. For an anchor $i \in \mathcal{I}$, the loss is given by

$$\mathcal{L}_i = \frac{-1}{|\mathcal{P}(i)|} \sum_{p \in \mathcal{P}(i)} \log \frac{\exp(\boldsymbol{z}_i \cdot \boldsymbol{z}_p/\tau)}{\sum_{a \in \mathcal{A}(i)} \exp(\boldsymbol{z}_i \cdot \boldsymbol{z}_a/\tau)}$$

By definition of the positive set $\mathcal{P}(i)$ in the multi-label supervised contrastive setting, if $\tilde{\boldsymbol{y}}_p = \tilde{\boldsymbol{y}}_i$, then $p \in \mathcal{P}(i)$, and thus $(i, p)$ constitutes a positive pair.

For any such $p$, the corresponding term in the loss is

$$-\log \frac{\exp(\boldsymbol{z}_i \cdot \boldsymbol{z}_p/\tau)}{\sum_{a \in \mathcal{A}(i)} \exp(\boldsymbol{z}_i \cdot \boldsymbol{z}_a/\tau)}.$$

Minimizing this term is equivalent to maximizing the numerator $\exp(\boldsymbol{z}_i \cdot \boldsymbol{z}_p/\tau)$, while relatively suppressing contributions from all other samples in the denominator. Therefore, the optimization objective enforces

$$\boldsymbol{z}_i \cdot \boldsymbol{z}_p \quad \text{to increase for all } p \in \mathcal{P}(i).$$

At the same time, for any $a \notin \mathcal{P}(i)$, the loss implicitly penalizes large values of $\boldsymbol{z}_i \cdot \boldsymbol{z}_a$, creating a separation between positive and non-positive samples.

At convergence (i.e., near a stationary point of $\mathcal{L}_i$), this yields

$$\boldsymbol{z}_i \cdot \boldsymbol{z}_p \gg \boldsymbol{z}_i \cdot \boldsymbol{z}_a, \quad \forall a \notin \mathcal{P}(i).$$

Assuming standard normalization of representations (i.e., $\|\boldsymbol{z}_i\| = \|\boldsymbol{z}_p\| = 1$), we have

$$\boldsymbol{z}_i \cdot \boldsymbol{z}_p \leq 1,$$

and the loss is minimized when

$$\boldsymbol{z}_i \cdot \boldsymbol{z}_p \to 1.$$

This implies that the angle between $z_i$ and $z_p$ tends to zero, and hence

$$\boldsymbol{z}_i \simeq \boldsymbol{z}_p.$$

Therefore, if $\tilde{\boldsymbol{y}}_p = \tilde{\boldsymbol{y}}_i$, their projected representations become arbitrarily close under the contrastive learning objective, which completes the proof.

$\square$

### B.2 Proof of Theorem 1

**Theorem 1.** *Let $\mathcal{K}_m^s$ and $\mathcal{K}_m^d$ be the Similarity and Dissimilarity operators, respectively, as defined in Eq. (14) and (15). For the case $m = 1$, their product vanishes:*

$$\mathcal{K}_m^s \mathcal{K}_m^d = 0, \quad \text{when } m = 1 \tag{19}$$

*Proof.* Consider the case where $m = 1$. By definition, we have $\mathcal{S} \cap \mathcal{T} = \varnothing$. This implies:

$$|\tilde{\boldsymbol{y}}_p^s| = |\mathcal{S} \cap \mathcal{T}| = |\varnothing| = 0$$

$$\therefore \mathcal{K}_1^s = \frac{|\tilde{\boldsymbol{y}}_p^s|}{|\tilde{\boldsymbol{y}}_i|} = \frac{0}{|\tilde{\boldsymbol{y}}_i|} = 0$$

Since $\mathcal{K}_1^s = 0$ and $\mathcal{K}_1^d$ is finite by construction, we conclude:

$$\mathcal{K}_1^s \mathcal{K}_1^d = 0 \cdot \mathcal{K}_1^d = 0 \tag{27}$$

$\square$

### B.3 Proof of Theorem 2

**Theorem 2.** *Consider the Similarity operator $\mathcal{K}_m^s$ and Dissimilarity operator $\mathcal{K}_m^d$ as defined in Eq. (14) and (15). For the case $m = 2$, their product equals unity:*

$$\mathcal{K}_m^s \mathcal{K}_m^d = 1, \quad \text{when } m = 2 \tag{20}$$

*Proof.* Consider the case where $m = 2$. By hypothesis, we have $\mathcal{S} = \mathcal{T}$. This equality implies:

$$\mathcal{K}_2^s = \frac{|\mathcal{S} \cap \mathcal{T}|}{|\mathcal{S}|} = \frac{|\mathcal{S}|}{|\mathcal{S}|} = 1$$

$$\mathcal{K}_2^d = \frac{1}{1 + |\mathcal{T} \setminus (\mathcal{S} \cap \mathcal{T})|} = \frac{1}{1 + |\varnothing|} = 1$$

where we have used the fact that $\mathcal{T} \setminus (\mathcal{S} \cap \mathcal{T}) = \varnothing$ when $\mathcal{S} = \mathcal{T}$. Thus, we conclude:

$$\mathcal{K}_2^s \mathcal{K}_2^d = 1 \cdot 1 = 1 \tag{28}$$

$\square$

### B.4 Proof of Theorem 3

**Theorem 3.** *Let $\mathcal{K}_m^s$ and $\mathcal{K}_m^d$ be the Similarity and Dissimilarity operators as defined in Eq. (14) and (15), respectively. For $m \in \{3, 4, 5\}$, their product is strictly bounded between 0 and 1:*

$$0 < \mathcal{K}_m^s \mathcal{K}_m^d < 1 \tag{21}$$

*Proof.* Consider $m \in \{3, 4, 5\}$. Under these cases, we have:

$$\mathcal{S} \cap \mathcal{T} \neq \varnothing \tag{29}$$
$$\mathcal{S} \neq \mathcal{T} \tag{30}$$

We first establish the strict positivity. Given $|\mathcal{S}| > 0$ and conditions 29-30, we have:

$$\mathcal{K}_m^s = \frac{|\mathcal{S} \cap \mathcal{T}|}{|\mathcal{S}|} > 0$$

$$\mathcal{K}_m^d = \frac{1}{1 + |\mathcal{T} \setminus (\mathcal{S} \cap \mathcal{T})|} > 0$$

For the upper bound, we consider three cases:

Case 1 ($m = 3$): By Equation 10, we have three conditions: $\mathcal{S} \cap \mathcal{T} \neq \varnothing$, $\mathcal{S} \not\subseteq \mathcal{T}$, and $\mathcal{T} \not\subseteq \mathcal{S}$. These conditions lead to:

$$|\mathcal{S} \cap \mathcal{T}| < |\mathcal{S}| \implies \mathcal{K}_3^s < 1$$
$$|\mathcal{T} \setminus (\mathcal{S} \cap \mathcal{T})| > 0 \implies \mathcal{K}_3^d < 1$$

Therefore, $\mathcal{K}_3^s \mathcal{K}_3^d < 1$.

Case 2 ($m = 4$): When $m = 4$, by Equation 11, we have $\mathcal{S} \supseteq \mathcal{T}$. This subset relation implies:

$$\mathcal{K}_4^s = \frac{|\mathcal{S} \cap \mathcal{T}|}{|\mathcal{S}|} = \frac{|\mathcal{T}|}{|\mathcal{S}|} < 1$$

$$\mathcal{K}_4^d = \frac{1}{1 + |\mathcal{T} \setminus (\mathcal{S} \cap \mathcal{T})|} = \frac{1}{1 + |\varnothing|} = 1$$

where the strict inequality $\mathcal{K}_4^s < 1$ follows from $|\mathcal{T}| < |\mathcal{S}|$ (since $\mathcal{S} \supsetneq \mathcal{T}$), and $\mathcal{K}_4^d = 1$ is a consequence of $\mathcal{T} \setminus (\mathcal{S} \cap \mathcal{T}) = \varnothing$ when $\mathcal{S} \supsetneq \mathcal{T}$. Therefore:

$$\mathcal{K}_4^s \mathcal{K}_4^d = \mathcal{K}_4^s \cdot 1 = \mathcal{K}_4^s < 1$$

Case 3 ($m = 5$): When $m = 5$, by Equation 12, we have $\mathcal{S} \subsetneq \mathcal{T}$. This subset relation implies:

$$\mathcal{K}_5^s = \frac{|\mathcal{S} \cap \mathcal{T}|}{|\mathcal{S}|} = \frac{|\mathcal{S}|}{|\mathcal{S}|} = 1$$

$$\mathcal{K}_5^d = \frac{1}{1 + |\mathcal{T} \setminus (\mathcal{S} \cap \mathcal{T})|} = \frac{1}{1 + |\mathcal{T} \setminus \mathcal{S}|} < 1$$

where $\mathcal{K}_5^s = 1$ follows from the fact that $\mathcal{S} \cap \mathcal{T} = \mathcal{S}$ when $\mathcal{S} \subsetneq \mathcal{T}$. The strict inequality $\mathcal{K}_5^d < 1$ holds because:

$$\mathcal{S} \subsetneq \mathcal{T} \implies |\mathcal{T} \setminus \mathcal{S}| > 0$$
$$\implies 1 + |\mathcal{T} \setminus \mathcal{S}| > 1$$
$$\implies \frac{1}{1 + |\mathcal{T} \setminus \mathcal{S}|} < 1$$

Therefore, we can conclude:

$$\mathcal{K}_5^s \mathcal{K}_5^d = 1 \cdot \mathcal{K}_5^d = \mathcal{K}_5^d < 1$$

Combining the results with Propositions 1 and 2, we obtain complete ordering for all $m \in \{1, 2, 3, 4, 5\}$. The products $\mathcal{K}_m^s \mathcal{K}_m^d$ satisfy:

$$0 = \mathcal{K}_1^s \mathcal{K}_1^d < \mathcal{K}_m^s \mathcal{K}_m^d < \mathcal{K}_2^s \mathcal{K}_2^d = 1, \quad m \in \{3, 4, 5\} \tag{31}$$

$\square$

## B.5 Proof of Theorem 4

**Theorem 4.** *Let $i \in \mathcal{I}$ be a fixed anchor sample, and let $p_3, p_4 \in \mathcal{P}(i)$ be positive samples corresponding to relations $R_3$ and $R_4$, respectively. Suppose their label spaces satisfy the cardinality constraint:*

$$|\tilde{\boldsymbol{y}}_{p_3}| = |\tilde{\boldsymbol{y}}_{p_4}| \tag{22}$$

*Then, the product of similarity and dissimilarity operators satisfies the strict inequality:*

$$\mathcal{K}_4^s \mathcal{K}_4^d > \mathcal{K}_3^s \mathcal{K}_3^d \tag{23}$$

*Proof.* Let us establish the strict inequality $\mathcal{K}_4^s \mathcal{K}_4^d > \mathcal{K}_3^s \mathcal{K}_3^d$ through direct comparison. From definitions 14 and 15, we have:

$$\mathcal{K}_4^s \mathcal{K}_4^d = \frac{|\tilde{\boldsymbol{y}}_{p_4}|}{|\tilde{\boldsymbol{y}}_i|} > \mathcal{K}_3^s \mathcal{K}_3^d = \frac{|\tilde{\boldsymbol{y}}_{p_3} - \tilde{\boldsymbol{y}}_{p_3}^d|}{|\tilde{\boldsymbol{y}}_i|} \cdot \frac{1}{1 + |\tilde{\boldsymbol{y}}_{p_3}^d|}$$

$$\Rightarrow$$

$$\frac{|\tilde{\boldsymbol{y}}_{p_4}|(1 + |\tilde{\boldsymbol{y}}_{p_3}^d|)}{|\tilde{\boldsymbol{y}}_i|(1 + |\tilde{\boldsymbol{y}}_{p_3}^d|)} > \frac{|\tilde{\boldsymbol{y}}_{p_4} - \tilde{\boldsymbol{y}}_{p_3}^d|}{|\tilde{\boldsymbol{y}}_i|(1 + |\tilde{\boldsymbol{y}}_{p_3}^d|)}$$

$$\Rightarrow$$

$$|\tilde{\boldsymbol{y}}_{p_4}|(1 + |\tilde{\boldsymbol{y}}_{p_3}^d|) > |\tilde{\boldsymbol{y}}_{p_3} - \tilde{\boldsymbol{y}}_{p_3}^d|$$

By the cardinality constraint 22 in the theorem:

$$|\tilde{\boldsymbol{y}}_{p_3}|(1 + |\tilde{\boldsymbol{y}}_{p_3}^d|) > |\tilde{\boldsymbol{y}}_{p_3} - \tilde{\boldsymbol{y}}_{p_3}^d|$$

where the strict inequality follows from the fact that for any positive real numbers $a, b > 0$:

$$a(1 + b) > a - b$$

This inequality holds trivially, thereby establishing the original claim $\mathcal{K}_4^s \mathcal{K}_4^d > \mathcal{K}_3^s \mathcal{K}_3^d$. $\square$

## B.6 Proof of Theorem 5

**Theorem 5.** *Let $i \in \mathcal{I}$ be a fixed anchor sample, and let $p_3, p_5 \in \mathcal{P}(i)$ be positive samples corresponding to relations $R_3$ and $R_5$, respectively. Suppose:*

$$|\tilde{\boldsymbol{y}}_{p_5}^d| \leq |\tilde{\boldsymbol{y}}_{p_3}^d| \tag{24}$$

*Then, the product of Similarity and Dissimilarity operators satisfies the strict inequality:*

$$\mathcal{K}_5^s \mathcal{K}_5^d > \mathcal{K}_3^s \mathcal{K}_3^d \tag{25}$$

*Proof.* From definitions 14 and 15, we have:

$$\mathcal{K}_3^s \mathcal{K}_3^d = \frac{|\tilde{\boldsymbol{y}}_{p_3} - \tilde{\boldsymbol{y}}_{p_3}^d|}{|\tilde{\boldsymbol{y}}_i|} \cdot \frac{1}{1 + |\tilde{\boldsymbol{y}}_{p_3}^d|}$$

$$\mathcal{K}_5^s \mathcal{K}_5^d = \frac{1}{1 + |\tilde{\boldsymbol{y}}_{p_5}^d|}$$

Taking the ratio:

$$\frac{\mathcal{K}_5^s \mathcal{K}_5^d}{\mathcal{K}_3^s \mathcal{K}_3^d} = \frac{|\tilde{\boldsymbol{y}}_i|(1 + |\tilde{\boldsymbol{y}}_{p_3}^d|)}{|\tilde{\boldsymbol{y}}_{p_3} - \tilde{\boldsymbol{y}}_{p_3}^d|(1 + |\tilde{\boldsymbol{y}}_{p_5}^d|)}$$

By the properties of cardinality and set difference:

$$|\tilde{\boldsymbol{y}}_{p_3} - \tilde{\boldsymbol{y}}_{p_3}^d| \leq |\tilde{\boldsymbol{y}}_i|$$

Given the constraint 24, $|\tilde{\boldsymbol{y}}_{p_5}^d| \leq |\tilde{\boldsymbol{y}}_{p_3}^d|$, we have:

$$\frac{|\tilde{\boldsymbol{y}}_i|(1 + |\tilde{\boldsymbol{y}}_{p_3}^d|)}{|\tilde{\boldsymbol{y}}_{p_3} - \tilde{\boldsymbol{y}}_{p_3}^d|(1 + |\tilde{\boldsymbol{y}}_{p_5}^d|)} > 1$$

Therefore, $\mathcal{K}_5^s \mathcal{K}_5^d > \mathcal{K}_3^s \mathcal{K}_3^d$. $\qquad\square$

## C  Experimental Setup

### C.1  Computing Cost Analysis

The key computation in all supervised contrastive loss functions, including ours, is the similarity between representations and the size of the positive set $\mathcal{P}(i)$ for each anchor $i$. All methods share the same core structure:

$$\mathcal{L}_i = \frac{-1}{|\mathcal{P}(i)|} \sum_{p \in \mathcal{P}(i)} \log \frac{\exp(\boldsymbol{z}_i \cdot \boldsymbol{z}_p / \tau)}{\sum_{a \in \mathcal{A}(i)} \exp(\boldsymbol{z}_i \cdot \boldsymbol{z}_a / \tau)}$$

Our method introduces a reweighting factor $\mathcal{K}_{i,p}^s \mathcal{K}_{i,p}^d \in [0, 1]$ only in the numerator of the positive terms:

$$\mathcal{L}_i^{\text{our}} = \frac{-1}{|\mathcal{P}(i)|} \sum_{p \in \mathcal{P}(i)} \log \frac{\mathcal{K}_{i,p}^s \mathcal{K}_{i,p}^d \exp(\boldsymbol{z}_i \cdot \boldsymbol{z}_p / \tau)}{\sum_{a \in \mathcal{A}(i)} \exp(\boldsymbol{z}_i \cdot \boldsymbol{z}_a / \tau)}$$

Importantly:

- The denominator (negative set) is unchanged across ALL, ANY, MulSupCon, and our method.

- The positive set $\mathcal{P}(i)$ for our method is structurally identical to ANY, typically larger than ALL but smaller or equal to MulSupCon, which forms multiple sets.

- The computation of $\mathcal{K}_{i,p}^s = \frac{|\mathcal{S} \cap \mathcal{T}|}{|\mathcal{S}|}$ and $\mathcal{K}_{i,p}^d = \frac{1}{1 + |\mathcal{T} \setminus (\mathcal{S} \cap \mathcal{T})|}$ involves simple set operations on one-hot label vectors, which are efficient and linear in the number of labels.

- No additional encoder forward passes or model parameters are introduced.

Empirically, memory and runtime cost for the Similarity-Dissimilarity Loss:

- Does not increase memory usage, as it does not alter the encoder architecture or increase feature dimensionality.

- Adds minimal runtime cost, primarily due to lightweight scalar computations per positive pair (i.e., computing $\mathcal{K}_{i,p}^s \mathcal{K}_{i,p}^d$).

In contrast, methods like MulSupCon may require repeated label-wise positive sets and multiple forward loss computations per label, depending on implementation. Thus, our method maintains comparable or better efficiency while offering a more principled weighting mechanism.

## C.2 Datasets and Metrics

To rigorously evaluate the efficacy of our proposed loss function, we conducted comprehensive experiments across three distinct data modalities: visual data, textual data, and specialized medical corpus data (MIMIC datasets). The MIMIC datasets are particularly noteworthy for their exceptionally large label space and pronounced long-tailed distributions (Huang et al., 2024a). This long-tailed characteristic, which is especially prevalent in multi-label classification scenarios, facilitates a robust assessment of the performance of our loss function across heterogeneous data distributions. Comprehensive statistical analyses of all experimental datasets are presented in Table 5.

- **MS-COCO** (Microsoft Common Objects in Context) (Lin et al., 2014) consists of over 330,000 images annotated across 80 object categories, providing rich semantic information for object detection, segmentation, and captioning tasks that has significantly advanced computer vision research since its introduction by Microsoft.

- **PASCAL VOC** (Everingham et al., 2010) contains 9,963 natural images with standardized annotations spanning 20 object categories, enabling rigorous evaluation of classification, detection, and segmentation algorithms in computer vision.

- **NUS-WIDE** (Chua et al., 2009) is a large-scale web image collection comprising approximately 269,000 Flickr images annotated with 81 concept categories and user tags, widely used as a benchmark for multi-label image classification.

- **AAPD** (Arxiv Academic Paper Dataset) (Yang et al., 2018) is a text corpus containing 55,840 scientific paper abstracts from arXiv with multi-label annotations across various subject categories, designed specifically for benchmarking multi-label text classification and document categorization algorithms.

- **MIMIC-III** [6] (Johnson et al., 2016) includes records labeled with expert-annotated ICD-9 codes, which identify diagnoses and procedures. We adhere to the same splits as in previous works (Mullenbach et al., 2018), employing two settings: MIMIC-III-Full, which includes all ICD-9 codes, and MIMIC-III-50, which includes only the 50 most frequent codes.

- **MIMIC-IV** [7] (Johnson et al., 2020) contains records annotated with both ICD-9 and ICD-10 codes, where each code is subdivided into sub-codes that often capture specific circumstantial details. we follow prior studies (Nguyen et al., 2023) and utilize four settings: MIMIC-IV-ICD9-Full, MIMIC-IV-ICD9-50, MIMIC-IV-ICD10-Full, and MIMIC-IV-ICD10-50.

**Metrics**. Consistent with prior research (Mullenbach et al., 2018; Nguyen et al., 2023), we report macro/micro-AUC, macro/micro-F1, and precision at K (P@$\mathcal{K}$) metrics on MIMIC datasets, where $\mathcal{K} = \{5, 8\}$ for different settings. Moreover, micro/macro-F1 and mAP are used for image datasets following (He et al., 2020; Zhang & Wu, 2024; Audibert et al., 2024).

## C.3 Baselines and Encoders

We compare the proposed Similarity-Dissimilarity Loss with representative supervised contrastive learning objectives for multi-label classification. These baselines cover different strategies for defining positive pairs and weighting multi-label relations.

- ALL method regards a sample as positive only when its label set is exactly identical to that of the anchor.

- ANY strategy relaxes the positive-pair definition by considering a sample positive if it shares at least one label with the anchor.

---

[6]We are granted access to MIMIC-III Clinical Database (v1.4)
[7]We are granted access to MIMIC-IV (v2.2)

Table 5: Statistics of datasets.

| Dataset | Train | Val | Test | Total # labels | Avg # labels |
|---|---|---|---|---|---|
| MS-COCO | 82.0k | 20.2k | 20.2k | 80 | 2.9 |
| PASCAL | 5.0k | 2.5k | 2.5k | 20 | 1.5 |
| NUS-WIDE | 125.4k | 41.9k | 41.9k | 81 | 2.4 |
| AAPD | 37.8k | 6.7k | 11.3k | 54 | 2.4 |
| MIMIC-III-Full | 47,723 | 1,631 | 3,372 | 8,692 | 15.7 |
| MIMIC-III-50 | 8,066 | 1,573 | 1,729 | 50 | 5.7 |
| MIMIC-IV-ICD9-Full | 188,533 | 7,110 | 13,709 | 11,145 | 13.4 |
| MIMIC-IV-ICD9-50 | 170,664 | 6,406 | 12,405 | 50 | 4.7 |
| MIMIC-IV-ICD10-Full | 110,442 | 4,017 | 7,851 | 25,230 | 16.1 |
| MIMIC-IV-ICD10-50 | 104,077 | 3,805 | 7,368 | 50 | 5.4 |

- MulSupCon (Zhang & Wu, 2024) considers each label independently and aggregates contrastive objectives across labels instead of forming a single positive set for each anchor. This design allows samples sharing multiple labels with the anchor to contribute more than samples sharing fewer labels.

- Jaccard (Lin et al., 2023) weights positive pairs according to the Jaccard similarity between their label sets.

- Class prototypes (Gupta et al., 2023) represents each class using a prototype and encourages sample representations to align with the prototypes associated with their labels. This method introduces class-level semantic anchors rather than relying solely on instance-to-instance contrast.

- MSC (Audibert et al., 2024) highlights the importance of considering the long-tailed distribution of data, addressing issues such as the "lack of positives" and the "attraction-repulsion imbalance" for multi-label scenarios.

- We include previous state-of-the-art medical methods as baselines (see Table 3).

For experimental evaluation, we employ modality-specific encoder architectures tailored to each data type. For image data, ResNet-50 (He et al., 2016) serves as the encoder architecture, consistent with established previous studies (He et al., 2020; Chen et al., 2020; Zhang & Wu, 2024), which is a main and common encoder in previous works. For textual data, we utilize pre-trained large language models (LLMs), specifically RoBERTa-base (Liu et al., 2019) and Llama-3.1-8B (Grattafiori et al., 2024) with Low-Rank Adaptation (LoRA) (Hu et al., 2022). Additionally, for the specialized task of ICD coding on MIMIC datasets, we implement PLM-ICD (Huang et al., 2022), a model specifically designed for ICD coding using LLMs.

### C.4 Implementation Details

Within the MSCL framework, we implement a two-phase training method as established by Khosla (Khosla et al., 2020): (i) encoder training, wherein the model learns to generate vector representations that maximize similarity between instances of the same class while distinguishing them from other classes; and (ii) classifier training, which utilizes the trained encoder and freeze it to train the classifier. All the experiments are conducted on $2 \times$ Nvidia A6000 48GB.

In the representation training, we use a standard cosine learning rate scheduler with a 0.05 warm-up period and set the temperature $\tau = 0.07$. The projection head comprises two MLP layers with ReLU activation function and employs contrastive loss function for the training, where the projected representation $z_k = Proj(Enc(\tilde{x}_k)) \in \mathbb{R}^{D_P}$. Here $h = Enc(\tilde{x}_k)$ denotes the encoded feature vectors and the projection dimension

$D_P = 256$. For subsequent classifier training, the projection head is removed, a linear layer is appended to the frozen encoder, and binary cross-entropy (BCE) loss is utilized for optimization.

For image data, we employ ResNet-50 using stochastic gradient descent (SGD) with momentum. The input images are set up at a resolution of $224 \times 224$ pixels. For text data, RoBERTa-base and Llama-3.1-8B serve as backbone encoders implemented via Huggingface platform (Wolf et al., 2020). RoBERTa configures with a dropout rate of 0.1 and AdamW optimizer with a weight decay of 0.01, exempting bias and LayerNorm from weight decay. Compared with full-parameter fine-tuning, we employ LoRA (Hu et al., 2022) to efficiently fine-tune large model Llama. LoRA configures with the low-rank dimension $r = 16$, scaling factor $\alpha = 32$ and dropout as 0.1. There is no KV cache to save memory during training. To enhance computational efficiency, BFloat16 precision is used for the training. The hyperparameters and detailed configuration are shown our code.

## D    More Detailed Results and Analysis

### D.1    Detailed Results and Analysis on Image

Table 1 reports that our proposed method outperforms baselines across almost metrics on the image benchmarks (MS-COCO, PASCAL, and NUS-WIDE) , with the sole exception of a lower mAP than Jaccard and MSC on PASCAL, demonstrating the effectiveness of explicitly modeling label information in MSCL.

Figure 2a illustrates the comparison between Similarity-Dissimilarity Loss and MulSupCon as measured by micro- and macro-F1 metrics. The results indicate that our method yields substantially greater improvements in macro-F1 compared to micro-F1 across all image datasets. Specifically, macro-F1 increases by 5.7% on MS-COCO and 5.8% on NUS-WIDE, whereas micro-F1 exhibits more modest improvements of 2.9% and 2.0%, respectively. Macro-F1 assigns equal importance to each class regardless of its frequency, rendering it particularly appropriate for evaluating performance on imbalanced datasets where minority class prediction accuracy is critical (LeCun et al., 2015; Zhang et al., 2023). In contrast, micro-F1 places more considerable weight on classes with more samples, making it more appropriate when larger classes should have a more potent influence on the overall score (LeCun et al., 2015; Lin et al., 2017). Multi-label classification inherently faces more pronounced challenges with long-tailed distributions than single-label classification due to exponential output space complexity, intricate label co-occurrence patterns, and high annotation costs (Zhang et al., 2023). The observed superior improvement in macro-F1 metrics provides compelling evidence that our method demonstrates exceptional efficacy in addressing long-tailed distribution challenges, a capability particularly crucial in multi-label scenarios.

However, on the PASCAL dataset, our method demonstrates mere marginal improvements, with gains of 0.88/0.84/0.17 in micro/macro-F1/mAP, respectively. This limited enhancement can be attributed to the structural characteristics of PASCAL, wherein the average number of labels per instance is approximately 1.5 (as detailed in Table 5), causing the task to approximate single-label classification, particularly when the batch size is limited (Khosla et al., 2020). Consequently, loss functions specifically designed for multi-label scenarios exert minimal influence on model performance under these conditions. As Audibert et al. (2024) have demonstrated, the cardinality of the label space constitutes a significant determinant of model efficacy within MSCL .

Furthermore, Figure 2b reveals that the standard deviation across four methods for PASCAL equals 0.58/0.64/0.27 in micro/macro-F1/mAP, which are considerably lower than the corresponding standard deviations observed for the MS-COCO and NUS-WIDE. This statistical finding suggests that the efficacy of specialized multi-label loss functions diminishes significantly when the average label cardinality per instance approaches 1 in MSCL. This finding further corroborates our theoretical analysis and hypothesis in the Section 2, wherein Similarity-Dissimilarity Loss degenerates to single-label scenarios (see Equation 17).

### D.2    Detailed Results and Analysis on Medical Domain

We extend and evaluate our method on the medical domain, specifically for ICD coding. The results in Tables 6 and 6 demonstrate that our proposed loss function consistently surpasses baselines across all metrics in a

comprehensive evaluation, considering: (i) Diverse data distribution: full setting (long-tailed distribution) and top-50 frequent labels setting; (ii) Model architectures: RoBERTa, LLaMA, and domain-specialized PLM-ICD; and (iii) ICD code versions: ICD-9 and ICD-10. The consistent performance improvements observed across these multidimensional evaluation criteria provide substantial empirical evidence for the efficacy and generalizability of our proposed approach.

In the full setting, macro-F1 performance exhibits considerably lower compared to micro-F1, whereas the top-50 setting achieves approximately equal macro and micro-F1 scores. This disparity indicates that extreme long-tailed distributions remain challenging for both the MSCL framework and our method, despite the improvements achieved.

Table 6 reports that our method achieves superior results on MIMIC-IV-ICD9-Full compared to MIMIC-III-Full, despite both datasets employing identical ICD-9 coding standards. This marked performance differential can be attributed primarily to the more extensive training corpus available in MIMIC-IV-ICD9-Full (see in Table 5). While MIMIC-IV-ICD10-Full similarly comprises a substantial volume of clinical data, its considerably expanded label taxonomy introduces increased representational sparsity and presents additional computational and methodological challenges (Nguyen et al., 2023). Moreover, the MIMIC-IV-ICD10-50 dataset demonstrates consistent performance metrics in this restricted setting, providing empirical evidence that label space dimensionality constitutes a critical determinant of model training efficacy.

Comparative analysis of model performance reveals that Llama significantly outperforms RoBERTa across evaluation metrics, a finding attributable to scaling laws of LLMs and the extensive knowledge and training corpus during the pre-training phase (Kaplan et al., 2020; Bahri et al., 2024; Huang et al., 2024b; 2025). Although LLMs demonstrate considerable efficacy in domain-specific applications (Nori et al., 2023), our results indicate that PLM-ICD consistently surpasses both RoBERTa and Llama across all experimental configurations. This hierarchical performance pattern aligns with theoretical expectations, as PLM-ICD incorporates architecture and training paradigms specifically optimized for automated ICD coding tasks (Huang et al., 2022). Despite the increasing generalization capabilities of foundation models in diverse applications, significant questions persist regarding their capacity to achieve state-of-the-art performance on highly specialized tasks, particularly within the medical domain, without substantial domain-specific training or parameter-efficient adaptation techniques (Saab et al., 2024). Contemporary research on foundation model applications in biomedical domain has predominantly relied on specialized adaptation methods tailored to specific domain requirements. The comparative advantages of domain-specific pre-training becomes particularly evident following the development of initial foundation model architectures, as exemplified by widely implemented medical models such as Med-PaLM (Singhal et al., 2023) and Med-Gemini(Saab et al., 2024).

Therefore, compared with the enhancements via the contrastive training phase, the intrinsic knowledge within LLMs contributes substantially more to ICD coding efficacy. In particular, domain-specific knowledge representations emerge as critical factors of LLMs performance in medical applications.

# E   Robustness Analysis

## E.1   Long-tail Frequency Bucket Analysis

To further analyze the behavior of the proposed Similarity-Dissimilarity Loss under long-tailed label distributions, we conduct a frequency bucket evaluation on MS-COCO. This analysis is motivated by the inherent imbalance in multi-label datasets, which typically contain a few frequent categories alongside a much larger number of infrequent ones. In such settings, high-quality positive-pair construction is critical because tail categories are highly sensitive to noisy or weak positive signals.

We construct the buckets based on the instance distribution of the MS-COCO training set. Specifically, all 80 categories are sorted in descending order of their training instance counts. We then compute the cumulative proportion of instances across categories and divide the label space into two groups:

Table 6: Results on MIMIC-III-Full, MIMIC-IV-ICD9-Full and MIMIC-IV-ICD10-Full test sets. The best scores among backbone encoder models are marked in bold.

| Method | MIMIC-III-Full | | | | | MIMIC-IV-ICD9-Full | | | | | MIMIC-IV-ICD10-Full | | | | |
| | AUC | | F1 | | P@8 | AUC | | F1 | | P@8 | AUC | | F1 | | P@8 |
| | Macro | Micro | Macro | Micro | | Macro | Micro | Macro | Micro | | Macro | Micro | Macro | Micro | |
|---|---|---|---|---|---|---|---|---|---|---|---|---|---|---|---|
| **RoBERTa** | | | | | | | | | | | | | | | |
| ALL | 89.87 | 95.83 | 7.94 | 53.08 | 71.06 | 93.04 | 98.57 | 11.76 | 57.73 | 64.75 | 89.46 | 98.19 | 4.23 | 53.82 | 64.17 |
| ANY | 88.15 | 94.18 | 7.13 | 51.35 | 68.92 | 92.86 | 98.14 | 11.17 | 57.42 | 64.48 | 89.09 | 98.07 | 4.02 | 52.28 | 62.65 |
| MulSupCon | 90.37 | 96.38 | 8.64 | 54.16 | 71.24 | 93.87 | 99.34 | 12.83 | 58.67 | 65.89 | 90.53 | 98.74 | 4.56 | 54.09 | 65.46 |
| Ours | **90.78** | **96.67** | **9.19** | **54.63** | **71.38** | **94.13** | **99.36** | **13.08** | **58.85** | **66.29** | **90.68** | **98.86** | **4.72** | **54.89** | **66.07** |
| **Llama** | | | | | | | | | | | | | | | |
| ALL | 91.27 | 96.94 | 8.38 | 54.75 | 72.63 | 94.52 | 98.93 | 12.34 | 58.97 | 66.35 | 90.78 | 98.57 | 4.53 | 54.98 | 65.31 |
| ANY | 90.64 | 96.38 | 7.82 | 53.97 | 71.85 | 94.19 | 98.74 | 11.93 | 58.68 | 65.92 | 90.36 | 98.32 | 4.37 | 54.24 | 64.78 |
| MulSupCon | 91.68 | 97.23 | 8.79 | 55.36 | 72.94 | 94.87 | 99.42 | 12.96 | 59.35 | 66.73 | 91.15 | 98.97 | 4.72 | 55.29 | 66.16 |
| Ours | **91.93** | **97.57** | **9.26** | **55.87** | **73.28** | **95.14** | **99.58** | **13.37** | **59.69** | **67.14** | **91.38** | **99.15** | **4.94** | **55.67** | **66.59** |
| **PLM-ICD** | | | | | | | | | | | | | | | |
| ALL | 92.58 | 98.69 | 10.73 | 60.06 | 76.84 | 96.95 | 99.28 | 14.18 | 62.83 | 70.53 | 91.87 | 98.79 | 4.83 | 57.36 | 69.29 |
| ANY | 91.09 | 97.36 | 9.24 | 58.87 | 75.38 | 95.85 | 98.17 | 12.64 | 61.82 | 69.58 | 90.54 | 97.72 | 4.54 | 55.86 | 68.17 |
| MulSupCon | 93.46 | 99.13 | 11.68 | 61.42 | 77.65 | 97.86 | 99.32 | 14.47 | 64.23 | 71.97 | 92.83 | 99.38 | 5.43 | 58.15 | 70.19 |
| Ours | **94.47** | **99.43** | **12.46** | **62.34** | **78.42** | **98.47** | **99.59** | **15.04** | **64.95** | **72.95** | **93.75** | **99.57** | **5.74** | **58.76** | **70.79** |

Table 7: Results on MIMIC-III-50, MIMIC-IV-ICD9-50 and MIMIC-IV-ICD10-50 test sets. The best scores among backbone encoder models are marked in bold.

| Method | MIMIC-III-50 | | | | | MIMIC-IV-ICD9-50 | | | | | MIMIC-IV-ICD10-50 | | | | |
| --- | --- | --- | --- | --- | --- | --- | --- | --- | --- | --- | --- | --- | --- | --- | --- |
| | AUC | | F1 | | P@5 | AUC | | F1 | | P@5 | AUC | | F1 | | P@5 |
| | Macro | Micro | Macro | Micro | | Macro | Micro | Macro | Micro | | Macro | Micro | Macro | Micro | |
| **RoBERTa** | | | | | | | | | | | | | | | |
| ALL | 87.73 | 90.57 | 57.38 | 61.84 | 61.29 | 93.84 | 94.46 | 67.63 | 72.24 | 60.92 | 91.43 | 93.52 | 64.86 | 67.65 | 60.07 |
| ANY | 87.36 | 89.42 | 56.25 | 60.83 | 60.32 | 93.37 | 93.73 | 67.39 | 71.97 | 60.24 | 90.06 | 92.03 | 64.09 | 66.54 | 58.08 |
| MulSupCon | 88.02 | 91.24 | 57.83 | 62.26 | 61.53 | 94.73 | 95.28 | 68.63 | 73.32 | 61.98 | 92.09 | 93.95 | 65.43 | 68.54 | 61.36 |
| Ours | **88.86** | **93.14** | **60.03** | **62.43** | **62.06** | **94.92** | **95.43** | **69.05** | **73.54** | **62.23** | **92.43** | **94.34** | **66.07** | **70.24** | **62.09** |
| **Llama** | | | | | | | | | | | | | | | |
| ALL | 88.93 | 91.67 | 60.32 | 64.58 | 62.87 | 94.32 | 95.28 | 69.18 | 73.72 | 61.42 | 92.35 | 94.57 | 66.38 | 69.83 | 61.75 |
| ANY | 88.57 | 91.09 | 59.72 | 64.03 | 62.19 | 94.05 | 94.85 | 68.79 | 73.19 | 61.07 | 91.89 | 93.97 | 65.82 | 69.09 | 61.12 |
| MulSupCon | 89.21 | 92.13 | 60.85 | 65.12 | 63.23 | 94.74 | 95.83 | 69.76 | 74.46 | 61.95 | 92.73 | 95.12 | 66.85 | 70.57 | 62.43 |
| Ours | **89.54** | **92.49** | **61.32** | **65.67** | **63.69** | **94.97** | **96.07** | **70.21** | **74.87** | **62.32** | **93.07** | **95.53** | **67.23** | **71.23** | **62.81** |
| **PLM-ICD** | | | | | | | | | | | | | | | |
| ALL | 90.13 | 93.02 | 65.18 | 69.43 | 65.26 | 95.18 | 96.42 | 71.31 | 75.83 | 62.45 | 93.53 | 95.97 | 68.96 | 73.14 | 64.52 |
| ANY | 89.03 | 92.07 | 63.73 | 68.14 | 63.84 | 93.73 | 95.34 | 70.23 | 74.43 | 61.42 | 92.27 | 94.42 | 67.95 | 71.83 | 63.17 |
| MulSupCon | 91.23 | 94.04 | 66.17 | 70.32 | 66.42 | 96.32 | 97.63 | 72.64 | 76.93 | 63.83 | 94.43 | 97.32 | 70.15 | 74.23 | 65.63 |
| Ours | **91.82** | **94.63** | **67.15** | **71.07** | **67.32** | **97.28** | **98.32** | **73.52** | **77.84** | **64.82** | **94.93** | **97.85** | **70.62** | **75.14** | **66.23** |

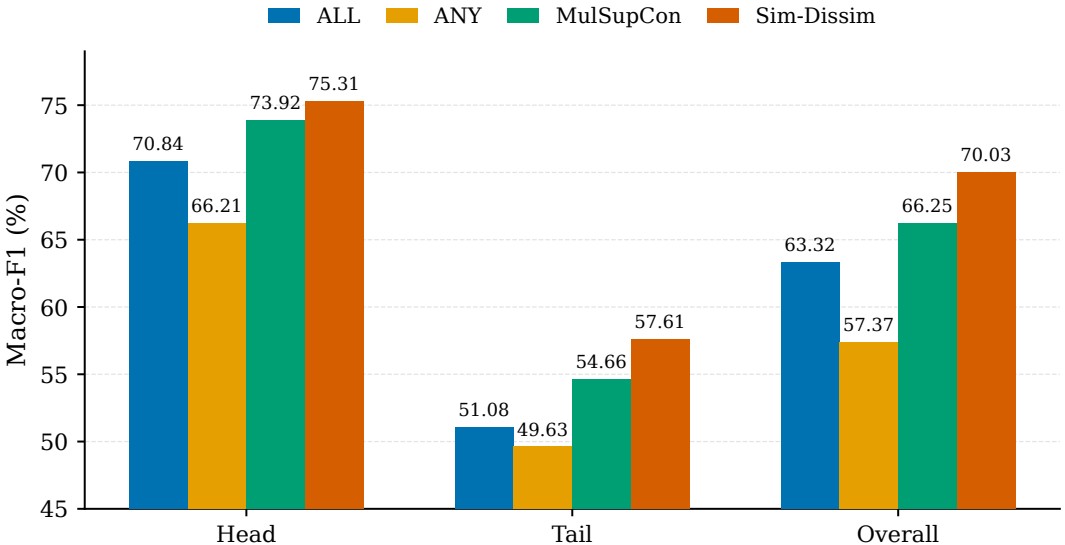

Figure 6: Long-tail frequency-bucket analysis on MS-COCO. The head bucket contains 23 classes accounting for the top 60% of training instances, and the tail bucket contains the remaining 57 classes.

- Head bucket: Contains the 23 categories that cumulatively account for the top 60% of training instances.

- Tail bucket: Contains the remaining 57 categories, which account for the remaining 40% of training instances.

This partition reflects the long-tailed nature of MS-COCO, where a small fraction of frequent categories dominate the training distribution. Finally, we evaluate macro-F1 separately on the head and tail buckets to compare the proposed method against ALL, ANY, and MulSupCon.

As shown in Figure 6, all methods perform better on head classes than on tail classes, as expected under long-tailed label distributions. Notably, the improvement of the proposed method is more pronounced on tail. Compared with MulSupCon, our method improves the macro-F1 by 1.39 on the head and by 2.95 on tail classes. This suggests that tail categories benefit more from the proposed relation-aware weighting mechanism.

These results are consistent with the motivation of the Similarity-Dissimilarity Loss. In long-tailed settings, tail labels have fewer positive instances and are therefore more vulnerable to ambiguous positive-pair assignments. Binary strategies such as ANY treat all overlapping samples uniformly, while MulSupCon mainly emphasizes shared labels without explicitly penalizing additional mismatched labels. In contrast, our method jointly considers shared and non-shared label information, thereby reducing the influence of weak or noisy positives. This leads to more reliable contrastive supervision for tail categories while preserving strong performance on head categories.

### E.2    Batch-size Sensitivity Analysis

We conduct a batch-size sensitivity analysis on the MS-COCO validation set to examine whether the proposed method remains effective under different numbers of in-batch samples. Since supervised contrastive learning are sensitive that performance can be affected by batch size (Khosla et al., 2020), in particular for the multi-label settings where positive pairs may have different degrees of label overlap.

We vary the batch size over {32, 64, 128, 256} and compare the proposed loss to ALL, ANY, MulSupCon. As shown in Figure 7, our method consistently outperforms baselines across all evaluated batch sizes on Micro-

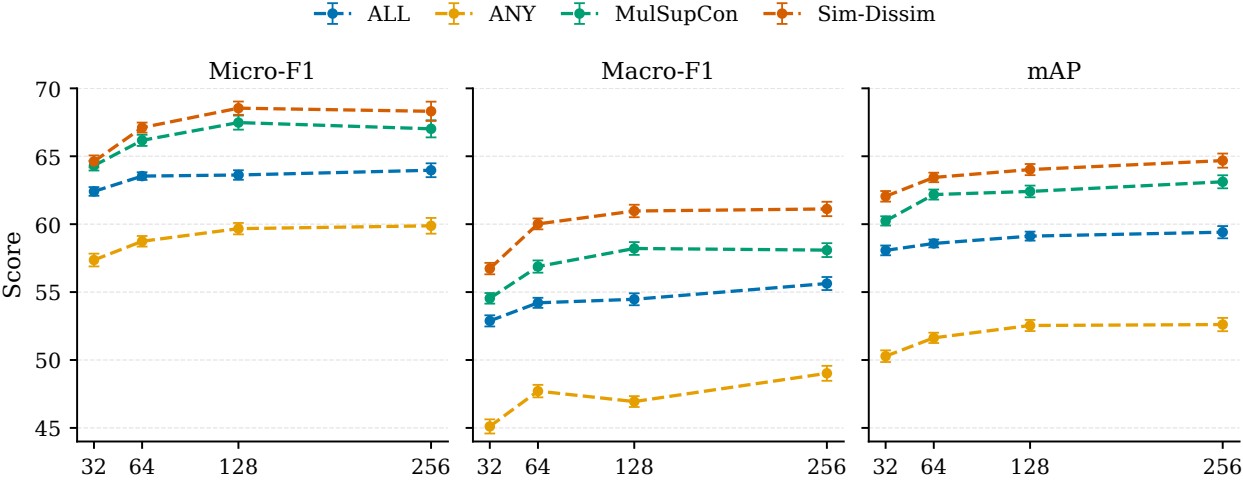

Figure 7: Batch-size sensitivity analysis on the MS-COCO validation set with batch sizes $\{32, 64, 128, 256\}$.

F1, Macro-F1, and mAP. This demonstrates that the proposed loss is not only effective under a specific batch configuration, but also robust across different levels of in-batch positive availability.

The advantage of our proposed loss is particularly clear for Macro-F1, where it maintains a stable margin over the baselines across all batch sizes. Since Macro-F1 is more sensitive to rare categories, this result further supports the effectiveness of the proposed similarity-dissimilarity weighting under imbalanced multi-label distributions. Overall, the results indicate that explicitly modeling both shared and mismatched label information provides consistently stronger contrastive supervision than strict, binary, or overlap-only positive-pair strategies.

# F  Additional Discussion of Related Work

Multi-label classification presents significant challenges due to its inherent label correlations, extreme and sparse label spaces, and long-tailed distributions. For instance, in the International Classification of Diseases (ICD) (Edin et al., 2023; Ji et al., 2024), the presence of one label (e.g., "Pneumococcal pneumonia") may increase the probability of co-occurring labels (e.g., "fever" or "cough"). Furthermore, multi-label datasets frequently exhibit long-tailed distributions, where a small subset of labels occurs with high frequency while the majority appear rarely. This imbalance typically results in models that perform adequately on common labels but underperform on infrequent ones (Zhang et al., 2023; Huang et al., 2023; Wang et al., 2024a). Additionally, the number of potential label combinations increases exponentially with the number of labels, resulting in heightened computational complexity and substantial memory requirements.

Contrastive learning aims to learn a representation of data such that similar instances are close together in the representation space, while dissimilar instances are far apart. Compared to self-supervised contrastive learning, such as SimCLR (Chen et al., 2020) and MoCo (He et al., 2020), Khosla et al. (2020) proposed supervised contrastive learning, which fully leverages class annotation information to enhance representations within the contrastive learning framework. Recent studies have extended supervised contrastive learning from single-label to multi-label scenarios by exploiting the additional information inherent in multi-label tasks. Zhang et al. (2022) proposed a hierarchical multi-label representation learning framework specifically designed to utilize comprehensive label information while preserving hierarchical inter-class relationships. Gupta et al. (2023) proposed a class-prototype-based supervised contrastive learning method for fine-grained multi-label educational video classification, where each class is represented by a learnable prototype and the objective encourages samples to be close to prototypes of their associated classes while being separated from prototypes of other classes.

In subsequent research, Zhang & Wu (2024) developed Multi-Label Supervised Contrastive Learning (Mul-SupCon), featuring a novel contrastive objective function that expands the positive sample set based on label overlap proportions. Similarly, the Jaccard Similarity Probability Contrastive Loss (JSPCL) (Lin et al., 2023) employed the Jaccard coefficient (Jaccard, 1912) to calculate label similarity between instances, sharing conceptual foundations with MulSupCon (Zhang & Wu, 2024) and MSC loss (Audibert et al., 2024) that those approaches primarily focus on similarity only, but ignoring dissimilarity.

Despite these advancements, the intricate relationships and dependencies between multi-label samples have yet to be fully elucidated. To address this gap, we introduce multi-label relations and formalize the concepts of similarity and dissimilarity. Inspired by the idea of re-weighting of logit adjustment (Menon et al., 2021), focal loss (Lin et al., 2017) and class-balanced loss (Cui et al., 2019), we leverage the similarity and dissimilarity factors to re-weight the contrastive loss, thereby enhancing discriminative power in multi-label scenarios.

## G    Limitations

Our experimental analysis on the PASCAL dataset reveals a crucial insight: the significant advantages of the proposed method appear primarily in multi-label dense scenarios. On PASCAL, where the average number of labels per instance is only approximately 1.5, the observed performance improvements are marginal compared to datasets such as MS-COCO or NUS-WIDE, which exhibit richer multi-label structures. This outcome is consistent with the theoretical underpinnings of our approach.

The limited gains can be attributed to the structural characteristics of PASCAL. With relatively few labels per instance, the task effectively reduces to a near single-label classification problem, particularly under constrained batch sizes. As demonstrated in Equation 17, under such conditions the proposed Similarity–Dissimilarity Loss degenerates to the standard supervised contrastive loss, thereby diminishing the benefit of re-weighting.

More broadly, in datasets with low label cardinality, the scarcity of relational diversity limits the expressiveness of our framework. Consequently, the advantages of dynamically re-weighted contrastive objectives become less pronounced, underscoring the dependence of our method on datasets that exhibit complex and diverse multi-label interactions.

