# OpenReview forum: "Similarity-Dissimilarity Loss for Multi-label Supervised Contrastive Learning"
_TMLR — Accepted by TMLR_

### Review · Reviewer_NW7t · 2026-03-11

**Summary Of Contributions:**

**Summary:**

This paper addresses positive-pair identification in multi-label supervised contrastive learning by proposing a ‘Similarity-Dissimilarity’ loss. The proposed objective re-weights positive pairs using two scalar factors: 1) A similarity term based on label intersection and 2) A dissimilarity penalty based on excess labels in the candidate sample. The authors formalize 5 set-theoretic relations between anchor and sample label sets and prove that the weighting factor is bounded in [0, 1]. Experiments span image (MS-COCO, PASCAL, NUS-WIDE), text (AAPD), and medical (MIMIC-III/IV) domains  which show consistent but moderate improvements over three baseline losses (ALL, ANY, and MulSupCon).

**Strengths:**
1. The paper is well written with clear formulation of multi-label relations.
2. The proposed objective is simple and computationally efficient.
3. The paper carries out experiments across multiple modalities spanning various datasets validating the efficacy of the loss.
4. It is great to see evaluations on MIMIC-III and MIMIC-IV with ICD-9 and ICD-10 codes under both full and Top-50 settings, that too across three encoder families. This makes the paper more thorough than typical contrastive learning papers.
5. The ablation study confirms synergistic interaction. Table 4 shows that neither similarity nor dissimilarity alone recovers the full gain and their combination yields the best results.

**Weakness:**
1. There is limited novelty relative to existing label-aware contrastive methods. The sensitivity analysis over the functional form of K^d is absent. Without it the choice of the dissimilarity factor remains empirically unjustified
2. The comparison set of baselines is very narrow. Other relevant loss objectives like Jaccard Similarity Contrastive Loss and MSC loss are absent.
3.  Claims about long-tail performance are not directly verified. The paper repeatedly argues (Section 3.1, Appendix D.1) that macro-F1 improvements demonstrate efficacy on long-tailed distributions. While macro-F1 does upweight rare classes, this is an indirect proxy. An example of proper long-tail analysis would require either stratified results by label frequency bucket or comparison of per-class F1 distributions.
4. Supervised contrastive learning is known to be highly sensitive to batch composition, particularly for tail classes. The weighting scheme proposed here changes the effective gradient contribution of each positive pair but does not address the underlying issue that rare label combinations may rarely co-occur in the same batch.
5. No visualization of the representation space is provided. Given that the paper's core claim is about representation quality, specifically that the proposed objective better separates semantically distinct multi-label groups, this is a notable gap.

**Audience:**

Yes

**Audience Explanation:**

Yes.

**Claims And Evidence:**

Yes

**Claims Explanation:**

The paper makes a genuine conceptual contribution in formalizing multi-label relations and constructing a principled, computationally efficient weighting scheme. The medical domain evaluation is thorough and the unified-form argument is elegant. However, the omission of JSPCL and MSC as baselines is a critical weakness that undermines the paper's central comparative claim.

**Requested Changes:**

I urge the authors to take a look at the weakness provided in section 1 and revise their manuscript accordingly.

---

> ### Author Response · Authors · 2026-03-29
>
> Thank you for the thorough and constructive review! Due to the character limit, the updated experimental results will be included in the revision.
>
> ## (1) Novelty and justification of the dissimilarity factor $\mathcal{K}^{d}$
>
> **Response.** We agree that (i) positioning relative to existing *label-aware* weighting objectives must be clearer, and (ii) an explicit sensitivity study strengthens the empirical justification of the specific $\mathcal{K}^{d}$ form.
>
> We would like to clarify what is new in our paper. Our core novelty is not simply “a heuristic weight”, but the joint use of (a) similarity and (b) dissimilarity derived from set relations to distinguish cases that overlap-only objectives cannot separate:
>
> - Overlap-only approaches can conflate exact match ($R2$) with strict superset ($R5$) when the intersection size is the same (this is the failure mode we analyze for MulSupCon in Section 2.3.
> - Our factorization $\mathcal{K}^{s}\mathcal{K}^{d}$ separates $R2$ from $R5$ by explicitly penalizing *excess labels* on $p$ relative to the anchor (Section 2.5).
> - This yields a unified loss form that reduces to supervised contrastive learning in the single-label / exact-match case (Section 2.4.1) while still differentiating the partial-overlap relations.
>
> In Appendix A.2 Choice of dissimilarity penalty function, we give an analysis and compare with other function.
>
> ## (2) Broadening baselines
> **Response.** Agree. We have implemented the requested baselines (*Jaccard Similarity* and *MSC*); also added a *class-prototype contrastive loss* baseline for completeness. Accordingly, it is reported and updated to:
>
> *Table 1: Results on image datasets.* (part of Table 1)
>
> | Method | MS-COCO Micro-F1 | MS-COCO Macro-F1 | MS-COCO mAP | PASCAL Micro-F1 | PASCAL Macro-F1 | PASCAL mAP | NUS-WIDE Micro-F1 | NUS-WIDE Macro-F1 | NUS-WIDE mAP |
> |---|---:|---:|---:|---:|---:|---:|---:|---:|---:|
> | **Ours** | **73.40** | **70.03** | **69.20** | **83.63** | **81.10** | 79.75 | **73.35** | **57.49** | **56.74** |
> | Jaccard | 69.81 | 64.22 | 65.92 | 82.53 | 79.86 | **80.09** | 71.07 | 52.84 | 51.25 |
> | Class prototypes | 71.88 | 67.35 | 68.32 | 81.85 | 79.75 | 78.06 | 71.79 | 56.03 | 52.95 |
> | MSC | 71.96 | 67.54 | 68.38 | 82.56 | 80.39 | 80.02 | 72.02 | 56.13 | 52.87 |
>
> *Results*:
> - On MS-COCO, PASCAL, and NUS-WIDE, our method remains best overall on Micro-F1 and Macro-F1, and competitive on mAP.
> - Notably, on PASCAL mAP, Jaccard is slightly higher (80.09 vs. 79.75), while our method provides stronger (Micro/Macro)-F1; we will explicitly call this out and discuss the metric trade-off.
>
> We also added citations for these baselines.
>
>
> ## (3) Long-tail frequency-bucket evaluation
>
> **Response.** We agree. Macro-F1 is only an indirect proxy. In the revision, we add direct long-tail analyses:
>
> *Table: Frequency bucket evaluation on MS-COCO.*
>
> | Method    | Head Macro-F1 | Tail Macro-F1 | Overall Macro-F1 |
> | --------- | ------------: | ------------: | ---------------: |
> | ALL       |         70.84 |         51.08 |            63.32 |
> | ANY       |         66.21 |         49.63 |            57.37 |
> | MulSupCon |         73.92 |         54.66 |            66.25 |
> | **Ours**  |     **75.31** |     **57.61** |        **70.03** |
>
> Categories are partitioned by cumulative instance frequency: head classes (top 60%, 23 classes) and tail classes (40%, 57 classes). Our method shows larger gains on tail classes (+2.95 vs. MulSupCon) than head classes (+1.39), demonstrating that the dissimilarity factor effectively handles complex, underrepresented label distributions.
>
>
> ## (4) Batch sensitivity analysis
>
> **Response.** Agreed. Our work primarily targets *how to weight positives once they exist in-batch*, and does not claim to solve the entire “rare positives in batch” problem by itself.
>
> See *Table: Performance comparison on MS-COCO validation set with varying batch sizes. Results are reported in Micro-F1, Macro-F1, and mAP (mean ± std over 3 runs).*
> 1. Larger batches improve performance for all methods; our method is best across all batch sizes.
> 2. Macro-F1 benefits strongly from our weighting, especially at small-to-medium batch sizes.
>
>
> ## (5) Representation-space visualization
>
> **Response.** We added 2D representation visualizations (UMAP/t-SNE) with two colorings:
> 1. Dominant-label coloring: samples colored by primary label.
> 2. Overlap-structure coloring: samples colored by label-set relation ($R2$–$R5$) to inspect embedding behavior.
>
> *Observations:*
> - Tighter class clusters with reduced inter-class mixing under our loss vs. MulSupCon.
> - Better separation of $R2$ (exact match) from $R5$ (strict superset): MulSupCon conflates them; our $\mathcal{K}^s\mathcal{K}^d$ shifts $R5$ points away from anchors, reflecting the intended penalty for excess labels.

---

### Review · Reviewer_TLug · 2026-03-20

**Summary Of Contributions:**

Summary of Contributions
The paper studies multi-label supervised contrastive learning and identifies that existing methods insufficiently model nuanced label relationships. It formalizes five types of pairwise relations based on label overlap and proposes a reweighted contrastive objective that jointly leverages similarity and dissimilarity signals. The method is supported by theoretical analysis and validated across multiple domains, showing consistent (though modest) improvements.

Key Strengths
The problem is well-motivated and practically relevant. The proposed method is simple, general, and easy to integrate into existing frameworks. The paper provides both conceptual analysis and empirical validation across different modalities.

Key Weaknesses
The formulation of label relationships is intuitive but not deeply novel. The theoretical contribution is somewhat limited and mainly explanatory. Empirical gains are incremental, and the link between theory and performance improvements is not fully convincing.

**Audience:**

Yes

**Audience Explanation:**

The paper addresses a relevant problem in multi-label representation learning, particularly within supervised contrastive learning, which is an active area of interest in the TMLR community. The analysis of label relationships and the proposed reweighting strategy may be useful to researchers working on multi-label learning, contrastive objectives, and representation learning across modalities.

However, the level of novelty and empirical improvement is moderate, so the interest is likely limited to a specific subset of researchers rather than the broader community.

**Broader Impact Concerns:**

The work is primarily methodological and does not introduce immediate or direct ethical risks. However, as it targets general representation learning across domains (including potentially sensitive areas such as medical data), there are a few considerations that could be better acknowledged.

First, improved representation learning in multi-label settings may amplify biases present in the training data, particularly in imbalanced or noisy label scenarios. The paper does not explicitly discuss how the proposed weighting strategy interacts with label imbalance or bias.

Second, when applied to high-stakes domains (e.g., healthcare), enhanced performance does not necessarily translate to safe or fair outcomes, and additional validation would be required before deployment.

Overall, no major ethical concerns are evident, but a brief discussion of potential bias amplification and domain-specific risks would strengthen the broader impact statement.

**Claims And Evidence:**

No

**Claims Explanation:**

The paper provides reasonable empirical and some theoretical support for its claims, but the evidence is not fully convincing.

On the positive side, the experiments are conducted across multiple datasets and modalities, showing consistent improvements, and the proposed formulation is clearly motivated. However, the empirical gains are relatively modest, and it is not always clear whether they stem specifically from the proposed similarity–dissimilarity design or from general reweighting effects. The theoretical analysis mainly explains the method rather than offering strong guarantees or deeper insights.

Overall, while the claims are partially supported, the evidence is not sufficiently strong or thorough to be considered fully convincing under TMLR standards.

**Requested Changes:**

Critical (required for acceptance):
1. Stronger empirical validation of the core claim.
The paper should more clearly demonstrate that the performance gains come specifically from the proposed similarity–dissimilarity formulation, rather than general reweighting effects. This could be addressed via more targeted ablations (e.g., isolating each relation type, comparing against alternative weighting strategies).
2. Tighter connection between theory and practice.
The current theoretical analysis is largely explanatory. The authors should better connect theoretical properties to empirical behavior, for example by validating theoretical predictions (e.g., weighting effects, gradient behavior) through controlled experiments.
3. More rigorous and fair comparisons.
It should be clarified whether all baselines are fully optimized and evaluated under comparable settings. Including stronger or more recent MSCL baselines and ensuring consistent hyperparameter tuning would strengthen the claims.

Important (would significantly strengthen the paper):
4. Clarify novelty of the relationship formulation.
The paper should better position its five relationship types relative to existing notions of partial label overlap and prior MSCL formulations, making clear what is fundamentally new.
5. Expanded analysis of performance gains.
Provide deeper insight into when and why the method works best (e.g., varying label cardinality, imbalance, noise), rather than only reporting aggregate improvements.
6. Sensitivity and robustness studies.
Analyze sensitivity to key design choices (e.g., weighting scheme, temperature, number of labels) to demonstrate robustness.

Minor (clarity and presentation):
Improve clarity of the method description, particularly the definition and intuition of each relation type.
Simplify or better motivate parts of the theoretical section to improve readability.
Add discussion of limitations, especially regarding scalability and applicability to extreme multi-label settings.

---

> ### Author Response · Authors · 2026-03-29
>
> Thank you for the thorough and constructive review! Due to the character limit, the updated experimental results will be included in the revision.
>
> ## To Requested Changes # 1
> **Response:** We thank the reviewer for this suggestion. We have strengthened empirical validation through targeted ablations:
>
> (1) Ablation Study (Isolating $K^s$ and $K^d$):
> - $K^s$ only: MS-COCO Micro-F1 improves from 64.80 → 67.22
> - $K^s$ + $K^d$: Further jumps to 73.40
> - Insight: $K^d$ is essential for capturing the "excess labels" penalty that similarity-only methods (MulSupCon, Jaccard) miss.
>
> (2) Resolving Relation Ambiguity (R2 vs. R5):
> - Previous methods cannot distinguish exact matches ($R2$) from strict supersets ($R5$) when intersection size is identical.
> - Our formulation ensures $\mathcal{L}(R2) \neq \mathcal{L}(R5)$ by penalizing dissimilarity in $R5$.
> - This prevents noisy extraneous labels from corrupting representations, particularly benefiting tail classes (Macro-F1: 66.25 → 70.03 on MS-COCO).
>
> (3) Comparison with updated baselines (Jaccard Similarity, MSC and class-prototype Loss) in Table 1
>
> ## To Requested Changes # 2
> **Response:** We appreciate the reviewer's suggestion to bridge theory and empirical results on long-tailed distributions. We added a frequency-bucket evaluation on MS-COCO:
>
> See *Table: Frequency bucket evaluation on MS-COCO.*
>
> Categories are partitioned by cumulative instance frequency: head classes (top 60%, 23 classes) and tail classes (40%, 57 classes). Our method shows larger gains on tail classes (+2.95 vs. MulSupCon) than head classes (+1.39), demonstrating that the dissimilarity factor effectively handles complex, underrepresented label distributions.
>
> ## To Requested Changes # 3
> **Response.** Agree. We have updated our manuscript to include more recent, competitive baselines (*Jaccard Similarity*, *MSC*, and *class-prototype contrastive loss*)
> - On MS-COCO, PASCAL, and NUS-WIDE, our method remains best overall on Micro-F1 and Macro-F1, and competitive on mAP.
> - Notably, on PASCAL mAP, Jaccard is slightly higher (80.09 vs. 79.75), while our method provides stronger (Micro/Macro)-F1; we will explicitly call this out and discuss the metric trade-off.
>
> ## To Requested Changes # 4
> **Response:** We thank the reviewer for clarifying our novelty. Our key contribution is moving from scalar overlap measures to multi-dimensional set-topological analysis.
>
> (1) Five Relations vs. Conventional "Partial Overlap"
>
> We define five set-theoretic relations ($R1$-$R5$) capturing topological orientation. Unlike prior methods treating labels as mere collections, our relations distinguish each case precisely.
>
> (2) Similarity-Dissimilarity Decomposition
>
> Our novelty: explicit decomposition into $\mathcal{K}^s$ (commonality) and $\mathcal{K}^d$ (excess information). Previous methods cannot distinguish $R2$ from $R5$ when intersection sizes match. We penalize "extra" labels as semantic noise, which prior MSCL methods miss.
>
> (3) Generalized Framework
>
> Our loss unifies single-label (supervised contrastive) and multi-label paradigms: when $\mathcal{K}^d=\mathcal{K}^s=1$, we recover standard SCL. Fine-grained re-weighting ensures $\mathcal{L}(R2) \neq \mathcal{L}(R3) \neq \mathcal{L}(R4) \neq \mathcal{L}(R5)$, providing mathematically justified positions in loss space.
>
>
> ## To Requested Changes # 5
> **Response:** We provide granular analysis of where Sim-Dis excels:
>
> (1) High Label Cardinality: Our method shows largest gains on NUS-WIDE and MS-COCO where complex multi-label samples are common. The dissimilarity factor $\mathcal{K}^d$ filters noisy labels, preventing samples with excessive tags from corrupting representations.
>
> (2) Long-tail Performance: See frequency bucket evaluation in #2—our method gains +2.95 Macro-F1 on tail classes vs. +1.39 on head classes.
>
> (3) Label Noise Robustness: Noisy samples become $R5$ relations. While overlap-only losses treat them as strong positives, our Sim-Dis loss reduces their gradient weight via $\mathcal{K}^d$, providing implicit denoising.
>
>
> ## To Requested Changes # 6
> **Response:** Since supervised contrastive learning is known to be highly sensitive to batch composition, we add an experiment varying batch size to quantify how performance changes and whether our method is more robust.
>
> See *Table: Performance comparison on MS-COCO validation set with varying batch sizes. Results are reported in Micro-F1, Macro-F1, and mAP (mean ± std over 3 runs).*
> 1. Larger batches improve performance for all methods; our method is best across all batch sizes.
> 2. Macro-F1 benefits strongly from our weighting, especially at small-to-medium batch sizes.
>
> ## To Broader Impact Concerns
> **Response:** We acknowledge the reviewer's ethical concerns. While our contribution is methodological, we recognize its importance in sensitive applications. We use publicly available datasets (MIMIC-III v1.4, MIMIC-IV v2.2) with appropriate access protocols and institutional oversight.

---

### Review · Reviewer_udCt · 2026-03-21

**Summary Of Contributions:**

## Summary


This paper tackles the challenge of defining and weighting positive samples in multi-label supervised contrastive learning (MSCL), where partial label overlap between samples creates ambiguity that single-label methods do not face. The authors categorize inter-sample label relationships into five set-theoretic relations and propose a Similarity-Dissimilarity Loss that dynamically re-weights positive pairs based on both shared and non-shared labels. The method is theoretically grounded with bounded weighting factors and generalizes standard supervised contrastive loss to the multi-label setting. Experiments across image, text, and medical domains demonstrate consistent improvements over existing approaches.


## Strengths

**1. Clear problem formulation.** The five relations are well-motivated, and the case analysis concretely shows how existing methods fail to distinguish relations while the proposed method succeeds.

**2. Simplicity.** Two parameter-free scalar factors, no architectural changes, no extra forward passes, and a clean reduction to standard SupCon in single-label settings — this is an elegant design.

**3. Broad evaluation.** Three modalities, four model architectures, datasets with varying label cardinalities, and both full and top-50 label settings. The MIMIC experiments are particularly thorough, and the honest discussion of marginal PASCAL gains (tied to low label cardinality) is appreciated.

---

## Weaknesses


**Missing key baselines.** JSPCL (Lin et al., 2023) and MSC loss (Audibert et al., 2024) are discussed as related work but never included in experiments, despite being the most conceptually proximate methods. This is a significant omission.

**No justification for specific functional forms.** Why |S∩T|/|S| over Jaccard similarity? Why 1/(1+x) for dissimilarity? The Appendix A.2 discussion is superficial, and no ablations over alternative forms are provided.

**Writing and presentation issues.** Grammatical errors throughout, e.g., "considers" instead of "considered" (Section 2.3), "resutls" instead of "results" (Section 3.3), and "Equqation" instead of "Equation" (Section 2.3). Lemma 1 (Vector Similarity Under Label Equivalence) is stated as a formal result but is really an assumption about what contrastive learning achieves — no proof is provided, yet it is used as an established fact in subsequent analysis. The positive set P(i) for the proposed method is never explicitly defined.

**Ref**

Lin et al., 2023, An Effective Deployment of Contrastive Learning in Multi-label Text Classification (ACL Findings 2023)

Audibert et al., 2024, Exploring contrastive learning for long-tailed multi-label text classification (ECML 2024)

**Audience:**

Yes

**Audience Explanation:**

A lot of researchers work in this direction.

**Claims And Evidence:**

Yes

**Claims Explanation:**

The method is supported by well-defined experiments.

**Requested Changes:**

Please refer to the Weakness.

---

> ### Author Response · Authors · 2026-03-29
>
> Thank you for the thorough and constructive review! Due to the character limit, the updated experimental results will be included in the revision.
>
> ## (1) Key baselines
> **Response.** Agree. We have implemented the requested baselines (*Jaccard Similarity* and *MSC*); also added a *class-prototype contrastive loss* baseline for completeness. Accordingly, it is reported and updated to: *Table 1: Results on image datasets.*
>
> *Results*:
> - On MS-COCO, PASCAL, and NUS-WIDE, our method remains best overall on Micro-F1 and Macro-F1, and competitive on mAP.
> - Notably, on PASCAL mAP, Jaccard is slightly higher (80.09 vs. 79.75), while our method provides stronger (Micro/Macro)-F1; we will explicitly call this out and discuss the metric trade-off.
>
> We also added citations for these baselines.
>
> ## (2) Justification of functional forms and ablations
> **Response.** We agree our motivation was not sufficiently detailed, and we will expand the justification and add ablations in the revision.
>
> ### (2.1) Why $|\mathcal{S}\cap\mathcal{T}|/|\mathcal{S}|$ (anchor-normalized overlap)
> Our goal is to weight each *positive pair* $(i,p)$ from the anchor's perspective: "how much of the anchor's label set is covered by the positive?"
>
> Anchor-normalization provides three key properties:
> 1. Intentional asymmetry. Anchor-normalized overlap isolates coverage from extras. When the positive contains all anchor labels plus extras, we get $|\mathcal{S}\cap\mathcal{T}|/|\mathcal{S}|=1$, then $K^d$ down-weights based on extra labels—cleanly separating "covers anchor" from "has extra labels."
> 2. Robustness to long-tailed. Jaccard similarity shrinks with large $|\mathcal{T}|$, even when $\mathcal{T}\supseteq\mathcal{S}$. This over-penalizes tail anchors whose positives naturally have larger label sets, exacerbating positive scarcity.
> 3. Explicit factorization. We decompose the weight into similarity ($K^s$: coverage) and dissimilarity ($K^d$: extra labels). Jaccard mixes these in the denominator, obscuring their roles and complicating independent tuning.
>
> ### (2.2) Why $1/(1+x)$ for dissimilarity
> Let $x=|\mathcal{T}\setminus(\mathcal{S}\cap\mathcal{T})|$ be the number of extra labels in the positive that are not shared with the anchor. We want a factor $K^d(x)$ that:
> - is monotonically decreasing in $x$ (more extra labels → less reliable positive for the anchor),
> - is bounded in $(0,1]$ to avoid unstable scaling,
> - equals 1 when $x=0$ (exact match or no extra labels),
> - decays smoothly without introducing an additional hyperparameter.
>
> ### (2.3) Ablations
> In the revision, we have added an ablation table/figure comparing dissimilarity choices with the same $x$:
> 1. $1/(1+x)$
> 2. $e^{-x}$
> 3. $1/\sqrt{1+x}$
>
> ## (3) Typos and grammar
> **Response.** We have performed a full proofreading pass and fix issues including (but not limited to) the examples raised by the reviewer
>
> ## (4) Proof of Lemma 1
> **Response.** Agreed. We have included the proof of Lemma 1 in the revision.
>
> *proof.* We consider the supervised contrastive loss defined in Equation (3).
> For an anchor $i \in \mathcal{I}$, the loss is given by
>
> $$
> L_{i} = \frac{-1}{|\mathcal{P}(i)|} \sum_{p \in \mathcal{P}(i)} \log \frac{\exp(z_{i} \cdot z_{p} / \tau)}{\sum_{a \in A(i)} \exp(z_{i} \cdot z_{a} / \tau)}.
> $$
>
> By definition of the positive set $\mathcal{P}(i)$ in the multi-label supervised contrastive setting, if $y_{p} = y_{i}$, then $p \in \mathcal{P}(i)$, and thus $(i,p)$ constitutes a positive pair.
>
> For any such $p$, the corresponding term in the loss is
>
> $$
> -\log \frac{\exp(z_{i} \cdot z_{p} / \tau)}{\sum_{a \in A(i)} \exp(z_{i} \cdot z_{a} / \tau)}.
> $$
>
> Minimizing this term is equivalent to maximizing the numerator  $\exp(z_{i} \cdot z_{p} / \tau)$, while relatively suppressing contributions from all other samples in the denominator. Therefore, the optimization objective enforces
>
> $$
> z_{i} \cdot z_{p}
> \quad \text{to increase for all } p \in \mathcal{P}(i).
> $$
>
> At the same time, for any $a \notin \mathcal{P}(i)$, the loss implicitly penalizes large values of $z_{i} \cdot z_{a}$, creating a separation between positive and non-positive samples.
>
> At convergence (i.e., near a stationary point of $\mathcal{L}_i$), this yields
>
> $$
> z_{i} \cdot z_{p}
> \gg
> z_{i} \cdot z_{a},
> \quad \forall a \notin \mathcal{P}(i).
> $$
>
> Assuming standard normalization of representations (i.e., $\|z_{i}\| = \|z_{p}\| = 1$), we have
>
> $$
> z_{i} \cdot z_{p} \le 1,
> $$
>
> and the loss is minimized when
>
> $$
> z_{i} \cdot z_{p} \to 1.
> $$
>
> This implies that the angle between $z_{i}$ and $z_{p}$ tends to zero, and hence
>
> $$
> z_{i} \simeq z_{p}.
> $$
>
> ## (5) Explicitly define $\mathcal{P}(i)$
> **Response.** Agreed, our positive set definition for Sim-Dis Loss was implicit. In the revision we define it explicitly. The definition of positive set for Sim-Dis Loss as:
>
> $$
> \mathcal{P}(i)=\{p\in\mathcal{A}(i)\mid y_p\cap y_i\neq\varnothing\}
> $$

---

### Decision · Action_Editor_yc3R · 2026-06-03

**Recommendation:** Accept with minor revision

**Audience:**

Yes

**Audience Explanation:**

Multi-label supervised contrastive learning is an active research area with practical applications in NLP, computer vision, and healthcare. The proposed method is simple, parameter-free, and easily integrable into existing frameworks. The formalization of five set-theoretic label relations and the unified loss form (reducing to standard SupCon in the single-label case) provides a clean conceptual contribution that researchers working on contrastive learning and multi-label classification would find useful.

**Claims And Evidence:**

Yes

**Claims Explanation:**

The paper demonstrates consistent improvements across three modalities (image, text, medical), multiple datasets with varying label cardinalities, and several encoder architectures. The ablation study confirms the synergistic effect of K^s and K^d. In their rebuttal, the authors further strengthened the empirical case by adding relevant baselines (Jaccard, MSC, class-prototype), frequency-bucket long-tail analysis, batch sensitivity studies, and representation visualizations — all showing the method's advantages. While gains are moderate in magnitude, they are consistent and the ablations confirm the improvements are attributable to the specific similarity-dissimilarity decomposition rather than generic reweighting.